# Sequence of abrupt transitions in Antarctic drainage basins before and during the Mid-Pleistocene Transition

Christian Wirths [1,2] ✉, Antoine Hermant[1,2], Christian Stepanek [3], Thomas F. Stocker[1,2] & Johannes C. R. Sutter[1,2]

Unraveling the drivers of climate variability during the Mid-Pleistocene Transition (MPT) remains a central challenge in paleoclimate research. This interval marked a shift from 41-kyr to 100-kyr glacial cycles associated with larger ice sheets. While previous studies emphasize interactions between climate and Northern Hemisphere ice sheets, Antarctica's role remains unclear. We use the Parallel Ice Sheet Model to simulate Antarctic Ice Sheet evolution over the last 3 million years, applying a climate index approach. Our simulations show that between 1.9 and 0.8 Ma, several Antarctic drainage basins underwent structural re-organization at different times, including the formation of a stable marine-based West Antarctic Ice Sheet (WAIS). We analyze the drivers of these thresholds and their associated state transitions. Our findings reveal tri-stability in the Thwaites basin and suggest that WAIS thresholds and their complex interactions amplified ~100-kyr climate variability before and during the MPT, providing new insights into long-term climate dynamics.

During the last 3 Myrs, global climate and its glacial-interglacial cycles have undergone substantial changes. While the global climate gradually cooled through the Pleistocene[1–3], the glacial-interglacial periodicity has shifted from a 41-kyr cycle to a 100-kyr cycle between 1.2 and 0.8 Ma BP, a period known as the Mid-Pleistocene Transition (MPT)[4,5] (see Fig. 1). Several hypotheses have been proposed to explain the MPT, from which many involve changes in Northern Hemisphere ice sheets. One prominent theory suggests that the gradual erosion of subglacial till sediment (regolith) over multiple pre-MPT glacial-interglacial cycles may have abruptly increased basal friction once the regolith has been fully removed[2,6]. This change could have facilitated the development of thicker, steeper Northern Hemispheric ice sheets, increasing resilience to obliquity-induced climate variations. Another hypothesis suggests that the gradual global cooling itself might have promoted larger Northern Hemispheric ice sheets, ultimately triggering a regime shift in ice sheet stability relative to obliquity-driven climate fluctuations[7]. A similar change in stability might have occurred in the Arctic sea-ice dynamics[8,9] in response to gradual global cooling. Finally, changes in the oceanic carbon cycle are suspected to have

occurred through a decoupling of Southern Ocean vertical mixing rates from sea surface temperature. This decoupling may have been triggered by changes in ice-ocean interactions and modifications to deep and bottom water formation processes[10].

Recent estimates by An et al.[11] suggest that Antarctic ice volume increased around 1.9 Ma BP, leading to additional global cooling through changes in wind and sea-ice patterns, ultimately enabling the formation of larger Northern Hemisphere ice sheets. Direct evidence of Antarctica's evolution during the Pleistocene is limited. However, ice-rafted debris (IRD) data from "Iceberg Alley" indicate increased activity within the Antarctic ice sheet (AIS) after 1.8 Ma BP[12] (see Fig. 1). Modeling studies have shown limited changes in the AIS prior to the MPT[13,14]. Nevertheless, the gradual decrease of southern deep ocean temperatures might have resulted in the formation of several marine-based ice margins during the mid-Pleistocene[15,16].

The AIS, particularly its predominantly marine-based western sector, is susceptible to rapid transitions and tipping behavior[17]. This vulnerability originates primarily from marine ice sheet instability[18,19] and potentially also marine ice cliff instability[19,20]. These instabilities, coupled

[1]Climate and Environmental Physics, University of Bern, Bern, Switzerland. [2]Oeschger Centre for Climate Change Research, University of Bern, Bern, Switzerland. [3]Alfred Wegener Institute, Helmholtz Center for Polar and Marine Research, Bremerhaven, Germany. ✉e-mail: christian.wirths@unibe.ch

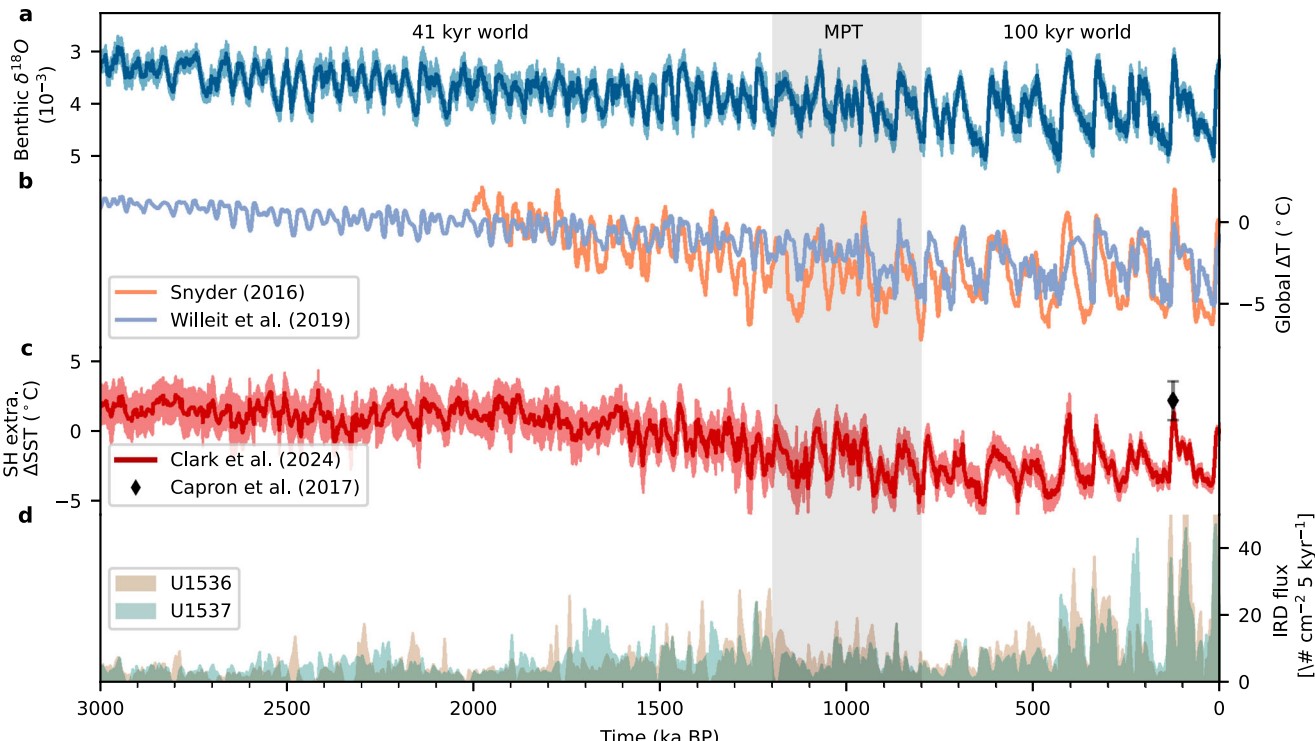

**Fig. 1 | Paleoclimate evolution over the last 3 million years. a** Global benthic $\delta^{18}O$ stack[62], **b** global near surface temperature anomalies from climate model simulations[2] and reconstructions[1], **c** Southern Hemisphere extra-tropical sea surface temperature (SST) anomalies[64,65] utilized in our simulations (see "Methods"), **d** ice rafted debris (IRD) flux from U1536 and U1537 in Dove Basin, southern Scotia Sea[12]. Grey shading marks the Mid-Pleistocene Transition (MPT).

with feedbacks involving bedrock rebound and ice surface elevation changes, cause the AIS to exhibit a hysteresis behavior under changing climatic conditions[13,21]. Consequently, future ice sheet evolution does not only depend on the future climatic drivers, but also on the history of the ice sheet state over several millennia before. Such a hysteresis behavior could profoundly influence how ice sheets respond to changing orbital and greenhouse gas conditions, affecting long-term climate cycles[7,22]. However, the regional ice dynamics of the AIS during the Quaternary remain largely unexplored. Especially, the catchment-specific evolution of the AIS and its dynamics have only been investigated for individual basins and shorter distinct time periods[23–26].

Here, we extend previous work to the whole time period since the late Pliocene and to all ice sheet basins of the AIS. Our study presents a three million year simulation of AIS evolution, resolving changes in individual drainage basins, as well as occurrence and timing of rapid transitions within these basins, with a special emphasis on the marine sections of the West Antarctic Ice Sheet (WAIS). Thereby, we present a continuous history of regional AIS dynamics across the Quaternary and their potential linkages to large-scale climate evolution.

## Results and discussion

### Three million years of Antarctic Ice Sheet variability

We use the Parallel Ice Sheet Model (PISM) to simulate the evolution of the AIS over the last three million years, employing a so called "climate index" approach[16]. This method combines individual climate snapshots from global climate models (GCMs) with a long-term climate evolution time series, derived from geological archives (hereafter referred to as the climate index), to force the ice sheet model (see "Methods" for details). Our ensemble of six simulations (see "Methods" and Table 1 for details) presented in this study suggest that Antarctic sea level equivalent ice volume was 8–9 m smaller during the late Pliocene than today, in agreement with proposed sea level ranges from previous studies[25,27,28]. Substantially warmer than present-day deep

**Table 1 | Chosen parameters for the six different ice sheet model configurations**

| Config. | $sia_e$ | $ssa_e$ | $\phi_{min}$ | $pQ$ | $\Delta SST_{target}^{LIG}[K]$ |
|---|---|---|---|---|---|
| 1 | 1.5 | 1.0 | 6° | 0.8 | 1.33 |
| 2 | 1.5 | 1.0 | 6° | 0.8 | 2.2 |
| 3 | 1.5 | 0.8 | 6° | 0.8 | 1.33 |
| 4 | 1.5 | 0.8 | 6° | 0.8 | 2.2 |
| 5 | 1.75 | 1.0 | 10° | 0.75 | 1.33 |
| 6 | 1.75 | 1.0 | 10° | 0.75 | 2.2 |

$sia_e$ and $ssa_e$ are the shallow ice and shallow shelf approximation enhancement factors utilized in the flow law. $\phi_{min}$ is the minimum till friction angle. $pQ$ is the pseudo plastic q exponent used in the sliding law relating shear stress and sliding velocities. Generally, a low q enhances sliding. $\Delta SST_{target}^{LIG}$ is the sea surface temperature (SST) target temperature applied in the bias correction of the oceanic forcing. The parameter combinations were selected by their capability to reproduce present-day observations of ice thickness, grounding line position and rate of ice volume change as in Wirths et al.[55].

ocean temperatures prevented the buildup of any large-scale ice shelves and kept the WAIS from advancing into marine basins (see Fig. 2). With the beginning of the Pleistocene (around 2.6 Ma BP), the climatic conditions episodically allowed glacial incursions in the Ross and Ronne basins, which however, consistently reversed during subsequent warmer interglacials. From around 2.0–1.9 Ma onward Antarctic ice volume showed a growth trend superimposed by glacial-interglacial variations, in accordance with recent proxy-based findings[11,12]. This long-term growth, primarily fueled by the expansion of the WAIS, lasted until the end of the MPT when the grounding line of Thwaites glacier advanced towards the present day ice margin (see Fig. S3 in the Supplementary Material). Relatively colder interglacial conditions prevailing between 800 ka and 400 ka contributed to the stabilization of the WAIS and maintained the total Antarctic ice volume

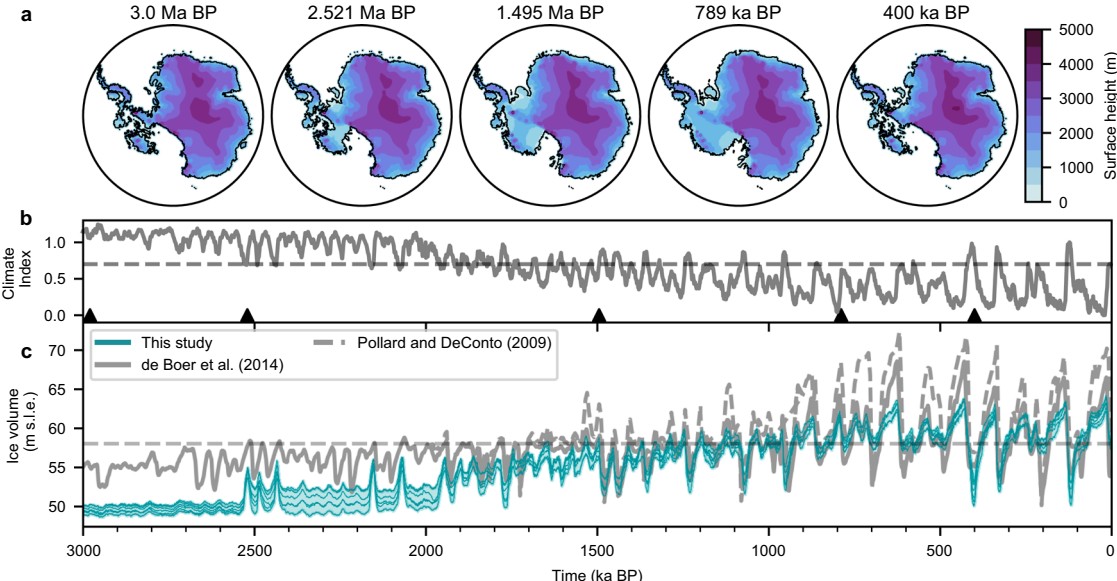

**Fig. 2 | Evolution of the Antarctic Ice Sheet and climate over the last 3 Myrs.**
**a** Simulated Antarctic grounded ice surface height and grounding line position of *Config. 1* (see Table 1). **b** Evolution of the applied climate index and **c** total sea level equivalent Antarctic ice volume of all ensemble members (see Table 1) together with results from Pollard and DeConto[13] and de Boer et al.[14]. Dashed thin lines indicate present day climate index and sea level equivalent ice volume values. Triangles mark time snapshots in (**a**).

above present-day levels until the Mid-Brunhes Transition. In agreement with IRD records[12], the subsequently warmer interglacial climate conditions during MIS 11, 9, and 5 led to destabilization and periodic collapse of the WAIS. In agreement with Golledge et al.[29,30] we find that, although the WAIS collapses during MIS5, the ice shelves within the Ronne basin remain intact.

## Different thresholds for regional variability changes

To disentangle the relationship between regional and large-scale patterns of ice sheet variability, we now focus on the simulated evolution of ice volume in individual drainage basins as defined by the IMBIE project[31]. The diverse responses in grounded ice volume across these basins are illustrated both as time series and phase space density against the climatic forcing index in Fig. 3 (selection) and Fig. S2 in the Supplementary Material (all basins). The phase space density quantifies the frequency of occurrence of a specific combination of ice volume and climate index through time, considering all six simulations. While we find that the largest West Antarctic basins (Thwaites (f), Ronne (g), and Ross (d)) exhibit several rapid transitions and thresholds over the last 3 Myrs, well aligned with the IRD record, other basins show a more gradual response towards changes in imposed climatic conditions. Based on the basin specific ice volume time-series, we introduce a classification of the Antarctic drainage basins into three distinct categories according to their response to climatic forcing. Type A basins (e.g., Wilkes, Princess Elisabeth, Enderby, and Kamp Land) exhibit a gradual response to climatic changes, suggesting that no major threshold for rapid transition has been crossed within the simulated climate conditions of the last 3 Myrs. Type B basins (e.g., Ross and Dronning Maud Land basin) demonstrate a substantial increase in grounded ice volume and associated variability once a specific forcing threshold is crossed. Type C basins (e.g., Thwaites, Amery, and Ronne basin), are characterized by two pronounced, rapid state transitions, indicating a more complex response to climate forcing. For these basins, the ice volume appears to transition between three preferred states, each showing limited response to climatic variations as long as no threshold is surpassed. This tri-state behavior is particularly evident for the Thwaites and Amery basins, as illustrated in Fig. 3 panels c) and e). The crossing of critical thresholds and the resulting state transitions can be associated with grounding line

advances and retreats in marine-based sections of individual Antarctic basins (see Fig. S3 in the Supplementary Material). While most of the West Antarctic basins are marine-based, East Antarctic basins generally have only limited marine-based sections (the Wilkes and Aurora basins being notable exceptions). This leads to significant differences between East and West Antarctica in terms of magnitude of transitions with respect to total ice volume. The Wilkes Basin, a predominantly marine-based region considered a potential past and future tipping element[17,32–34], stands as an exception in our simulations, having not crossed any substantial thresholds over the last 3 Myrs. However, modeling evidence for a Plio-Pleistocene collapse of the Wilkes basin remains inconclusive[25,35–37], heavily depending on the applied forcing, resolution, and ice sheet model parameterization[34,36]. The model resolution applied in this study is comparably coarse owing to the significant computational expense imposed by the long-term simulations that we conduct, and major ice streams and topographic troughs[32,38] of the Wilkes Subglacial basin may not be resolved in detail, hence our simulations not reproducing a similar instability (see Supplementary Material for details).

## Interconnected state transitions in the WAIS

To validate and quantify the thresholds that trigger substantial reorganizations within individual AIS basins and associated reversibility, as well as to investigate whether differences in the timing of climate changes and ice sheet responses could have led to additional dynamics beyond simple equilibrium behavior, we performed a series of hysteresis experiments. Specifically, we conducted a stepwise quasi-equilibrium hysteresis simulation, incrementally adjusting the climatic forcing index by 0.05 every 20 kyrs in both directions (cooling, warming), for all six model configurations. The resulting hysteresis curves of all model configurations robustly support our classification and reveal distinct thresholds and rapid, substantial state transitions, especially for the marine basins of the WAIS (see Figs. 4 and S1 in the Supplementary Material). For these basins, the crossing of thresholds is associated with significant changes in grounding line position, as also observed in our dynamic long-term simulation depicting the formation of the WAIS. Starting from a warm climate state with a largely ice free WAIS, the Ross and Ronne basins are the first to cross their specific threshold for ice sheet advance for an index

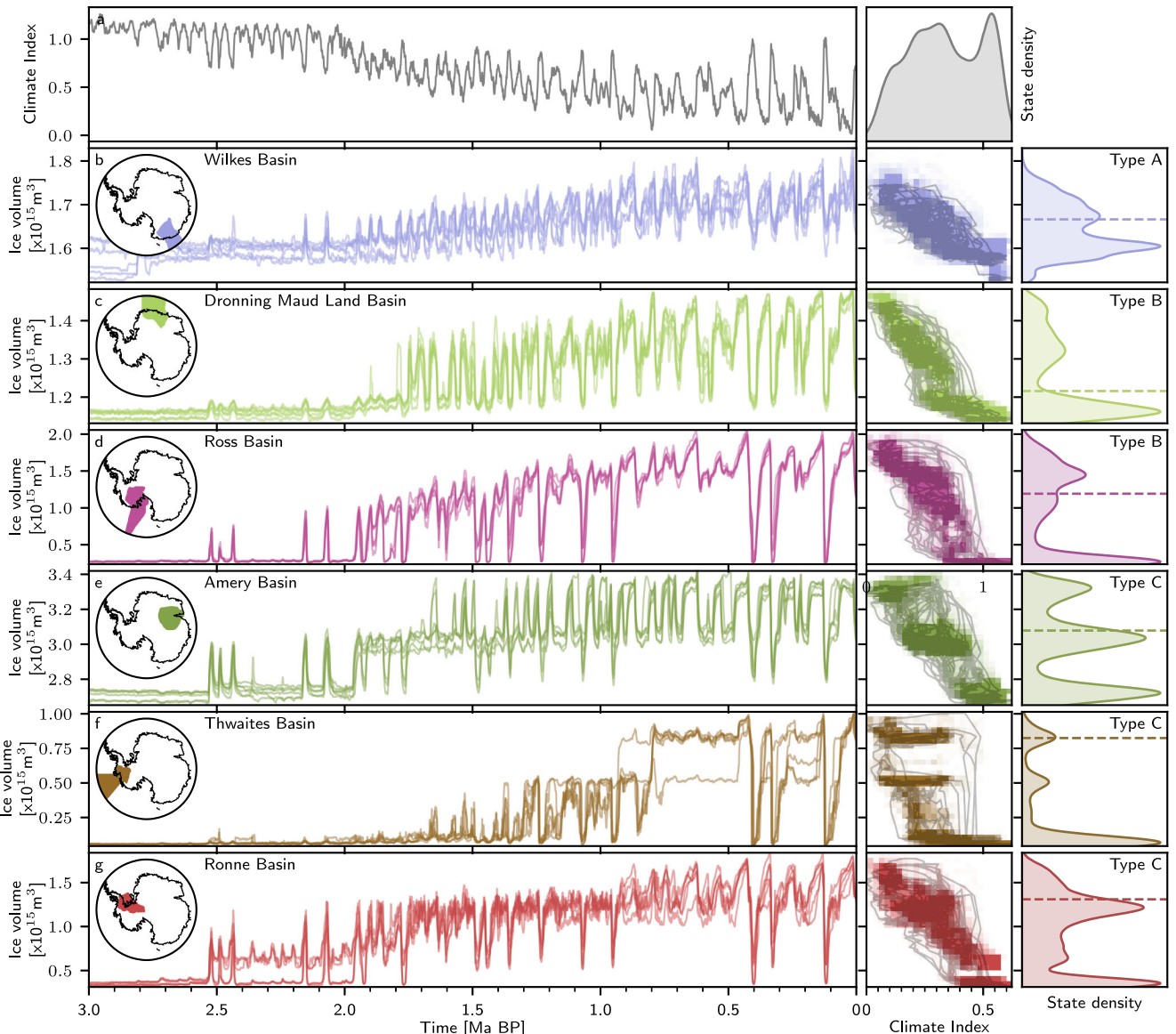

**Fig. 3 | Evolution of ice volume in individual Antarctic basins. a** Climate index against time (left column) and distribution (right column). Grounded ice volume (left column) of all six model configurations for the **b** Wilkes, **c** Amery, **d** Dronning Maud, **e** Thwaites, **f** Ronne, and (**g**) Ross basin. Average phase space distribution of the grounded ice volume (middle column) of all six model configurations as well as the phase space trajectory for *Config. 1* (see Table 1). Distribution of simulated grounded ice volume and type of dynamical behavior during the past 3 Ma (right column). Dashed lines in the right column indicate the mean grounded ice volume at the end of the simulation.

smaller than 0.8 and 0.85, aligning with the observed simulated periods of ice growth in these basins during the early Pleistocene. Our findings suggest that in a future scenario where WAIS has collapsed, even climate conditions marginally warmer than present-day could support the regrowth of certain WAIS sections. This potential regrowth, combined with similar behavior observed in some East Antarctic basins, could account for up to approximately 4.3 m of sea level equivalent ice volume. A second threshold for a rapid transition in West Antarctic ice volume is crossed for a climatic index below 0.55. Such climatic conditions support the glaciation of large sections of the Thwaites basin and additional ice volume growth in the Ross basin. When the forcing index surpasses below a threshold of 0.2–0.15, Thwaites grounding line advances further towards present-day/LGM conditions as it has happened around 800 ka BP in our transient simulations. Simultaneously, we simulate another episode of ice volume growth in the Ronne basin.

As the climate index reverses back toward warmer conditions, the Thwaites basin experiences an initial, minor drop in ice volume and retreating grounding line position, followed by a stable configuration across a wide range of climatic conditions up to, and beyond, present-day levels. However, when the climate index reaches a range between 0.75 and 0.9, the Thwaites as well as Ross basin cross another threshold, resulting in their complete collapse. Although these transitions occur at the same thresholds, our dynamic simulations reveal a sequence: initial ice loss in the Ross basin is followed by rapid grounding line retreat and ice loss in the Thwaites basin, ultimately causing the Ross basin to lose nearly all of its marine-based sections. Similar to findings by Pollard and DeConto[13], the overall WAIS ice volume hysteresis exhibits a flat intermediate section. However, our simulations show a more pronounced hysteresis behavior, which could be partly attributed to lower overall ice volume during glacial conditions.

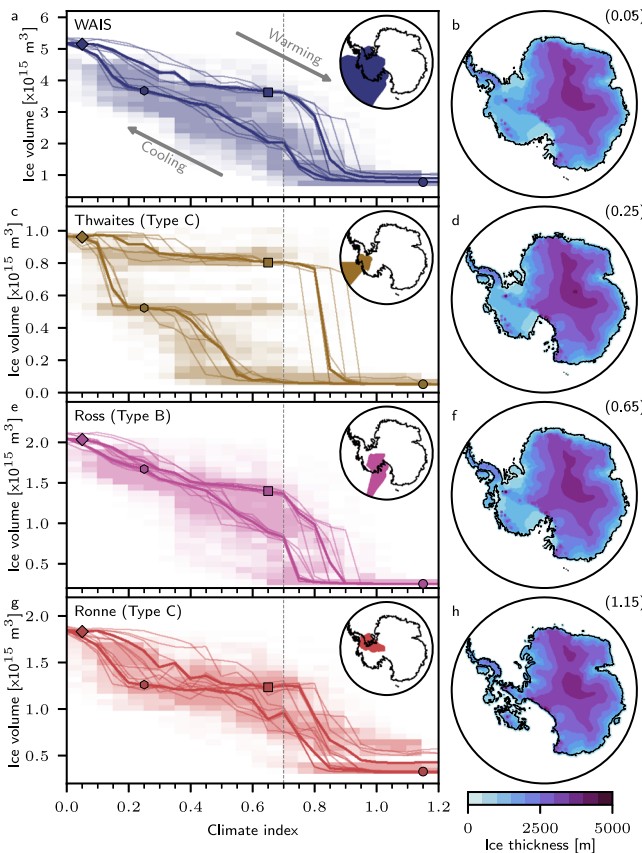

**Fig. 4 | Hysteresis of Antarctic grounded ice volume. a** West Antarctic Ice Sheet (WAIS) and its major basins: **c** Thwaites, **e** Ross, and **g** Ronne for climate conditions between the mPWP and LGM together with mean phase space density (shading) from all six transient 3 yrs simulations (25 bins). Grounded ice surface height (blue shading) and grounding line position (black line) and respective climate index value (in brackets) representing (**b**, diamond) the glacial state, (**d**, square) Holocene state, (**f**, hexagon) partially advanced WAIS state, and (**h**, circle) a fully collapsed WAIS state. Thin solid lines represent the six individual model configurations. Thick solid lines represent the median across all configurations. The vertical dashed gray line marks the present-day reference forcing.

Hysteresis behavior is evident in many Antarctic basins, but it is most pronounced in the tri-stable Thwaites basin. Notably, the intermediate ice volume state exhibits a high phase space density during our transient simulations (Fig. 4c). This phenomenon can only be partially explained by a hysteresis effect, as demonstrated by our reversal experiments on the intermediate state (see Fig. S4 in the Supplementary Material). These findings suggest that the intermediate state of Thwaites basin can be resilient against overshooting its equilibrium state threshold. This is a characteristic behavior for slow-onset tipping elements like ice sheets[39,40].

### 100 kyr-paced WAIS variability preceding the MPT
The temporal evolution of grounded ice volume reveals 80–120 kyr cycles for the Ronne and Ross basin from 1.9 Ma BP onwards and for the Thwaites Basin from 1.5 Ma BP onwards (see Fig. 3). In agreement with the time-series, the evolutionary spectrum of grounded ice volume variability (see Fig. S5 in the Supplementary Material) reveals that the highly nonlinear behavior of major West Antarctic sectors appears to substantially enhance 80–120 kyr and longer-term variability before and during the MPT. While geological time-scale variability might arise due to the long-term growth trend of Antarctic basins and as overtones of orbital-paced variability, we compare the power of 41 kyr to 100 kyr-paced variability to investigate the characteristic response of WAIS to those frequencies (see

Fig. 5). 41 kyr variability is the dominant signal in the climate index forcing until 1.2 Ma BP. However, 80–120 kyr variability dominates grounded ice volume variability in the Thwaites basin since the first emergence of ice in this area around 1.5 Ma BP. The cause of this enhancement seems to lie in the unique thresholds and their stability of the WAIS basins that might lead to the skipping of interglacial ice sheet collapses for certain ice sheet configurations and climatic conditions, which is discussed in more detail in the Supplementary material. Results similar to Thwaites, but less pronounced, are observed for the Ross and Ronne basins. Notably, 41 kyr variability is dominant or at par with 80–120 kyr variability in the Ronne and Ross basins in our simulations for most periods prior to the emergence of ice in the Thwaites basin, in agreement with the ANDRILL AND-1B sediment record[41], suggesting that the interconnections of these basins with Thwaites might be an explanation for the dominant 80–120 kyr-paced variability after 1.5 Ma BP. Due to effects on ocean circulation and carbon cycle via changes in Antarctic Bottom Water formation[42–44], Antarctica therefore may have enhanced 80–120 kyr glacial-interglacial climate variability prior to the MPT. This hypothesis would have to be put to the test in a fully coupled modelling framework, which is currently computationally prohibitive at the necessary spatial resolution and time span to be simulated.

Further, the model resolution of 16 km together with uncertainty within ice sheet model parametrization lead to an underestimation of the simulated present-day ice thickness for the WAIS when compared to observations (see Fig. S6 in the Supplementary Material). This could affect the stability of the WAIS and its intrinsic thresholds, as well as the pacing of glacial advances and retreats overall. Similarly, the lower than observed present-day velocities (see Fig. S6 in the Supplementary Material) within the Ronne and western Ross ice shelf might bias the stability of the grounding line within this area. In addition, uncertainties within the applied forcing snapshots, especially the warm ocean temperatures for Pliocene conditions and parameterization of basal melt rates (see Figs. S11 and S12 in the Supplementary Material), could lead to biases within the simulation, limiting WAIS advances during climatic periods warmer than the LIG.

Overall, we demonstrate a complex interplay between different regions of the AIS during the last 3 million years, translating climatic forcing into ice sheet volume variability. Prominently, the WAIS crosses several critical thresholds sequentially leading to the formation of a stable marine-based ice sheet. This sequence of events unfolds between 1.9 Ma and 0.8 Ma BP. Additionally, we present a tri-stable regime of the Thwaites basin, which amplified variability in the 80–120 kyr band up to and during the MPT. Our work supports the hypothesis that AIS dynamics may have contributed to the MPT via the formation of marine ice sheets and the associated climate feedbacks.

## Methods
### Ice Sheet model
To simulate the evolution of the AIS across the Pleistocene, we utilize the thermodynamically-coupled PISM version 2.1[45,46]. The basal yield stress is calculated following the Mohr-Coulomb law[47] with the till friction angle being determined by bed topography following a linear interpolation[46,48,49]. Ice flux across the calving front is calculated by applying the physically-based 2D-calving parameterization known as Eigen calving[50] together with thickness calving for ice shelves thinner than 75 m. Sub-shelf melt and refreezing is calculated from given salinity and temperature fields using the ocean box model PICO[51]. At the ice sheet base, we prescribe constant geothermal heat flux from Shapiro and Ritzwoller[52]. Further, we calculate the surface mass balance from precipitation and surface temperature using a positive degree day model[53,54]. We additionally employ an idealized cosine seasonal temperature cycle. To account for changes in ice sheet

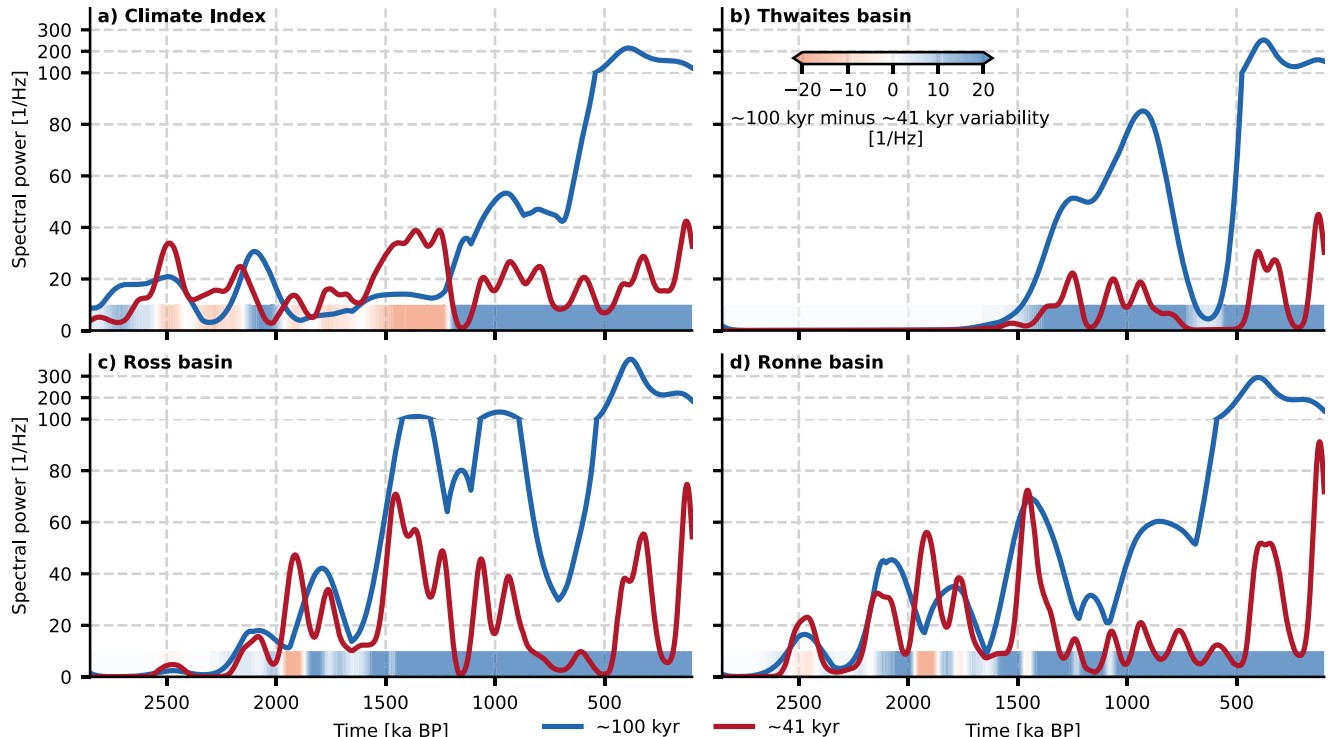

**Fig. 5 | Temporal evolution of the spectral power.** Maximum spectral power within the ~100 kyr (80-120 kyr) and ~41 kyr (39-43) kyr band for the **a** climate index, and grounded ice volume in **b** Thwaites, **c** Ross, and **d** Ronne basin. The bottom color shadings indicate the difference in spectral power between the ~100 kyr and ~41 kyr variability. While ~41 kyr variability is the dominant mode within the forcing index between 2.0 and 1.2 Ma BP, the West Antarctic Ice Sheet basins exhibit amplified dominant ~100 kyr variability before and during the Mid-Pleistocene Transition. Please note that before the spectral analysis, all time series were standardized, so the units of the spectral power correspond only to the inverse of the frequency.

elevation, we adjust the near-surface air temperature with the dry-adiabatic lapse rate (9.8 K/km) and the simulated thickness change. Following this thickness change-induced temperature variation, we exponentially scale precipitation with temperature by a scaling factor of $C = 0.07\,K^{-1}$. We apply a six-member parameter suite of varying ice flow parameters (see Table 1) to represent uncertainties with respect to basal boundary conditions and ice flow. The parameter combinations were selected by their capability to reproduce present-day observations of ice thickness, grounding line position, and rate of ice volume change after 30,000 years under constant present day forcing as in Wirths et al.[55].

## Climate index

PISM is transiently forced by precipitation, annual mean, and January mean near-surface air temperature and deep ocean (300–700 m) salinity and temperature calculated by interpolating between Last Glacial Maximum (LGM), Last Interglacial (LIG), and mid-Piacenzian Warm Period (mPWP) snapshots from the global circulation model Community Earth System Models (COSMOS)[56–59] using a climate index method[16]. At grid box location (i,j), a forcing variable at time $t$ can be calculated from the COSMOS snapshots by

$$T^{i,j}(t) = T_{ref}^{i,j} + \sum_{x\in\{LGM, LIG, mPWP\}} \omega_x(t)\Delta T_x^{i,j}, \qquad (1)$$

with reference forcing $T_{ref}$, weights $\omega_x$, and forcing anomalies $\Delta T_x^{i,j}$. For the LGM and LIG, the anomalies are calculated with respect to the COSMOS pre-industrial reference period, while for the mPWP the anomaly is calculated with respect to the LIG to produce a smooth and seamless transition. As reference forcing $T_{ref}^{i,j}$ we here use temporal average from 1979 to 2016 from the RACMO2.3p3[60] for the atmosphere and the ISMIP6[61] ocean forcing. The weights $\omega_x$ are calculated

from the climate index $CI$ with respective reference values for the different time periods ($CI_{LGM} = 0.0$, $CI_{LIG} = 1.0$, $CI_{mPWP} = 1.2$, and $CI_{ref} = 0.7$) by

$$\begin{aligned}
\omega_{LGM} &= 1.0 - \frac{\min(CI, CI_{ref})}{CI_{ref}}, \\
\omega_{LIG} &= \min\left(\frac{\max(CI, CI_{ref})-CI_{ref}}{CI_{LIG}-CI_{ref}}, 1\right), \\
\omega_{mPWP} &= \frac{\max(CI, CI_{LIG})-CI_{LIG}}{CI_{mPWP}-CI_{LIG}}.
\end{aligned} \qquad (2)$$

In summary, this procedure generates a climate forcing field for a given climate index by interpolating between the two adjacent climate model snapshots. To calculate precipitation, we apply exponential scaling of the temperature anomaly as described above. The climate index time-series applied in this study is based on a combination of global temperature reconstructions[1], benthic oxygen isotopes[62] and EPICA Dome C ice core records[63], following Sutter et al.[16].

## Climate forcing data

In this study, we use snapshot paleoclimate simulations performed with the COSMOS[56] for the LGM (LGM-ctl)[57], LIG (LIG-ctl)[58], and mPWP (Eoi400)[59]. From the post spin-up state of all GCM time slice simulations, we calculated the temporal mean of the annual and January near-surface air temperature, precipitation, as well as depth-integrated (300–700 m) ocean temperature and salinity. All fields were bi-linearly interpolated to the 16 km PISM domain over Antarctica. Then anomalies to the reference period were calculated by subtracting the COSMOS pre-industrial control simulation (E280)[59]. For the mPWP, we additionally accounted for the difference in ice sheet topography, for which we corrected by applying a uniform dry-adiabatic lapse rate to the topography difference with respect to the control simulation.

The oceanic forcing was extended over the entire Antarctic domain by a nearest neighbor extrapolation on a basin by basin level. Our basins are adapted from IMBIE[31] and can be found in Fig. S13 in the supplementary material. To avoid the extrapolation of small scale artifacts we convoluted ocean temperature and salinity with a two-dimensional Tophat kernel of radius 40 grid boxes (320 km) before the extrapolation step.

To better align our LIG and mPWP ocean temperature with SST reconstructions, we perform a bias correction. For this purpose, we establish a transfer function between the mean SST ($\Delta SST$) and depth-integrated temperature anomaly ($\Delta\theta$). For a given target SST anomaly $\Delta SST_{target}$ obtained from proxy reconstructions, we then calculate the bias-corrected depth-integrated ocean temperature by:

$$\theta^{i,j}_{corrected} = \theta^{i,j} + \frac{\Delta\theta}{\Delta SST}\left(\Delta SST_{target} - \Delta SST\right) \qquad (3)$$

In this study, we apply two target SST anomalies for the LIG (1.33 K[64] and 2.2 K[65]) and one for the mPWP (3.01 K[64]). Additionally, we apply a transient sea level forcing derived from the benthic $\delta^{18}O$ stack[62] using a linear $\delta^{18}O$-to-sea level calibration[66] anchored to LGM and LIG (see Supplementary Material and Fig. S14).

### Model initialization and spin-up

We initialize the model using present-day ice sheet and bedrock topography[67], followed by a two-stage spin-up process. The first stage involves running the model for 200 kyrs under transient climate conditions from 3.3 to 3.1 Ma BP, using parameter configuration *Config. 1* (see Table 1). In the second stage, we simulate the time period from 3.1 to 3.0 Ma BP for each individual parameter configuration, using the output from stage one as the initial condition. This methodology ensures a sufficient temperature profile within the ice sheet and allows the ice sheet to adapt to the substantially warmer climate conditions during the mPWP. The resulting initial ice sheet height and grounding line extent for our three-million-year-long simulations are shown for *Config. 1* in Fig. 2 as well as Fig. S5.

## Data availability

The ice volume, grounding line, and ice thickness data generated, analyzed, and presented in this study, have been deposited in the Zenodo database under the accession code https://doi.org/10.5281/zenodo.17279526[68].

## Code availability

PISM is openly available from https://github.com/pism/pismand Zenodo https://doi.org/10.5281/zenodo.17279526[68], and for this study version 2.1 was used. The code used to generate the main figures is has been deposited in the above-mentioned Zenodo repository (https://doi.org/10.5281/zenodo.17279526[68]).

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

## Acknowledgements

Calculations were performed on UBELIX, the high-performance computing cluster at the University of Bern. C.W. acknowledges funding by the Swiss National Science Foundation through the pleistoCEP2 project (grant no. 200492). C.W., J.C.R.S., and A.H. acknowledge funding from the Swiss National Science Foundation (grant no. 211542). J.C.R.S. acknowledges support from the European Union's Horizon Europe research and innovation program under grant agreement number 101137601 (ClimTip). This is ClimTip contribution number 121. T.F.S. acknowledges funding from the Swiss National Science Foundation (grant no. 200492) and from the European Union's Horizon 2020 Research and Innovation Program under grant agreement no. 820970

(project TiPES). C.S. acknowledges institutional funding from the Alfred Wegener Institute (AWI) via the research programme "Changing Earth - Sustaining our Future" of the Helmholtz Association as well as financial support from the Helmholtz Climate Initiative REKLIM and through the ERC grant "i2B" (grant no. 101118519), funded by the European Union.

## Author contributions

C.W., T.F.S., and J.C.R.S. devised the study. C.W., A.H., and J.S. implemented the climate index in PISM. C.S. provided the climate forcing data. C.W. set up the ice-sheet model, preprocessed the forcing, ran the ensemble simulations, and performed the analysis. C.W. led the writing of the paper with contributions from A.H., C.S., T.F.S., and J.C.R.S.

## Competing interests

The authors declare no competing interests.
