## [Transparent Peer Review file · Nature Communications]

Sequence of abrupt transitions in Antarctic drainage basins before and during the Mid-Pleistocene Transition

Corresponding Author: Dr Christian Wirths

Version 0:

Reviewer comments:

Reviewer #1

(Remarks to the Author)

In this manuscript entitled, "Cascade of abrupt transitions in Antarctic drainage basins before and during the Mid-Pleistocene Transition" the authors aim to understand main drivers of Quaternary climate variability during the Mid-Pleistocene Transition (MPT) through the dynamics of the Pleistocene Antarctic ice sheet using regional Parallel Ice Sheet Models. The results indicate that the eccentricity cycle was particularly prominent in the Thwaites basin, West Antarctic Ice Sheet (WAIS) before the MPT, providing new insights into the distinctive behavior of the marine-based WAIS in response to climate fluctuation contributed to the MPT. This study improves the resolution of the history and dynamics of the Antarctic Ice Sheet and is expected to have a substantial impact on various research areas related to the global climate system.

Overall, this paper is well designed and the data are presented logically, but I have some questions and comments as described below.

- Indeed, there are limited constraining data indicating Pleistocene Antarctic ice sheet fluctuations, but some geological reports exist (such as surface exposure ages). For example, observational data from DML suggest that around 3 million years ago, the inland region of East Antarctica experienced an increase in ice sheet thickness. Subsequently, as the global cooling, the ice sheet elevation decreased, indicating a change in water transport and ice sheet morphology (Yamane et al., 2015, Nature Communications). How do you interpret the consistency between these findings and this paper?
- Line 59: From Fig. 2, it appears that this study presents lower values around 3 Ma to 2 Ma compared to de Boer et al. (2024). Is 'agreement' the appropriate term here?
- Lines 91–97: The absence of thresholds in the Wilkes Basin seems significant and provides various implications. While I understand that this may be due to topographic resolution (model resolution), it might be helpful to include additional figures illustrating these effects.
- Ultimately, why does the Thwaites Basin ice sheet fluctuate with the eccentricity cycle? Additionally, does this mechanism differ from the variations in the Northern Hemisphere (Abe-Ouchi et al., 2013)? While I understand that complex interactions are involved, more detailed explanation be required. For instance, to what extent is the GIA component considered? What about dynamic elements? If the term 'dynamics' is used, the text include more discussion on factors such as flow velocity and ice sheet morphology.

Reviewer #2

(Remarks to the Author)

Wirths et al. NCOMMS

The paper presents a suite of long (3 My) transiently-forced ice sheet model experiments that simulate the evolution of the Antarctic Ice Sheet through the end-Pliocene and whole of the Pleistocene. From these model runs the authors extract ice volume timeseries for each discrete ice sheet drainage basin, and use the patterns in these timeseries to infer distinct 'types' of catchment. By reversing their experiments, the authors also explore hysteresis, and by using spectral analysis they make

connections between modelled transient behaviour and Milankovitch cyclicities.

The paper is generally well-written and the figures are of a very high quality. The simulations are impressive – doing long transient simulations such as these is not an easy undertaking. And the ideas presented are interesting and certainly add something to the discussion around the nature of the Mid-Pleistocene Transition and how the AIS responded at this time. Hysteresis experiments (long, whole continent) and catchment-scale AIS modelling (shorter, regional scale) have been done for several decades now, and there are a lot of papers on these topics (although not many are referenced, see below). The advance here I think is doing both these things together – very long simulations analysed at the catchment scale. So it is a valuable contribution in that sense. I do have some reservations about the way that some of the arguments are constructed however, and outline these below.

General comments

1) To me it seems that there are three parts to this paper: i) the identification of catchment-specific behaviours, ii) modelling hysteresis, and iii) orbital forcing. Of these, I found the first to be most compelling. I like the basin-level insights. But it isn't made clear what the hysteresis experiments actually add to the story, other than showing that the same series of events plays out in reverse. The orbital inferences I think are very weak, and this aspect either needs to be much more rigorously and thoroughly explored, or left out completely (see detailed comments below).

2) Evidence for cascades: to me, there is no evidence presented anywhere in the paper that supports the idea of a 'cascade' (in the sense that events in one basin trigger events in another, that then trigger another and so on). I expand on this below, but given this fundamental issue, I think the title and emphasis of the paper needs to be addressed. What is being shown is a sequential response of regionally-abstracted metrics to the imposed continent-wide forcing.

3) Importance of model setup: like most simulations that employ a spinup rather than data assimilation, the model used here doesn't agree perfectly with observations – it is too thin in WAIS and the ice shelves flow too slowly. That's totally fine, but I think it is important to have some discussion in the main text on how this affects the results. Similarly, see the final comment below on 3-2 Ma ocean temperature variability in the applied forcing – the peaks are very warm and most likely prohibit GL advance for WAIS. The implications of this need to be explained in the main text.

4) The referencing is ok, but misses a lot of other work that is relevant. For example, I counted only about a dozen of the > 50 references in the main paper that are AIS modelling studies – the rest are mostly climate modelling, proxy, or theoretical / methodological studies. Given that the focus of the manuscript is on modelling the AIS, and that some novelty is required to justify publication, I would suggest that a more in-depth reading of the literature might be useful to show a) what else has been done previously regarding AIS catchment types and catchment-scale AIS tipping points / transitions, and b) what the new study adds to that.

5) There are quite a few typos and grammatical errors (especially in the SI material) – I haven't detailed them below and so I would suggest a careful line-edit to find and address these.

Specific comments

Title: Might need to revisit this in light of comments below about lack of evidence for 'cascades'

Line 12-13: 'eccentricity-paced' – I have some issues with this, see later comments.

Line 17: In addition to Berends, perhaps it would be worth reading the more recent and comprehensive paper by Herbert 2023.

Fig. 1 and elsewhere. Might be worth mentioning that Ahn et al 2017 effectively supersedes LR04

Line 61: I think the onset of the Pleistocene is closer to 2.6 Ma (2.58 Ma according to the International Strat. Commission).

Line 74: I think 'phase space density' needs to be introduced and explained, because it is not an intuitive metric, and a lot of the discussion hangs on this.

Line 78: How exactly are these types identified? Is it simply a visual grouping based on perceived similarity? Or is it quantified in some way? I think it needs to be defined and justified. The problem I think is that grouping based on the timeseries appearance is very subjective, and grouping by state density histogram shape could yield different results. For example, in Fig. 3, Thwaites histogram looks more like a Type B than a Type C, whilst Ross (Type B) actually also looks quite like Ronne (Type C). Perhaps some objective approach can be found to distinguish between the catchments?

Line 84: 'three preferred states' – this isn't clear for Ronne, and the histogram really only shows two. Again, I would suggest finding a way to formalise the way these states are defined.

Line 121-123: This is really the only place in the manuscript which talks about the so-called 'cascade', but to me it is simply a sequential pattern of change – there is no mechanistic or causal link demonstrated. The problem I think is that both sectors (Ross and Thwaites) are being forced synchronously with the same timeseries perturbation, so it is natural that they will both

respond to that forcing in a way that is conditioned by their geometry & dynamic configuration. To show a causal link, you'd probably need to have an experiment where the forcing was only applied in one of these sectors, to see whether changes in A lead to changes in B, which trigger further changes in A etc. It might be easier to just avoid saying there is a cascade...

Line 126-131: It seems that this paragraph is the only text to explore and justify the hysteresis experiments, but I still don't really understand what these 3 sentences in particular are saying: "Notably, the intermediate ice volume state exhibits a high phase space density during our transient simulations (Fig. 4 c). This phenomenon can only be partially explained by a hysteresis effect, as demonstrated by our reversal experiments on the intermediate state (see. Fig. A4). These findings suggest that the intermediate state of Thwaites basin can be resilient against overshooting its equilibrium state threshold." Can this be clarified, or just removed?

Fig. 5 – Are there units for the colour bar? I'm not sure what these 'standardized' numbers mean.

Fig 5 caption – it is fine to state the window of the different spectral bands (eg 80-120, 39-43), but it is not clear that they can be unequivocally linked to eccentricity and obliquity respectively. Huybers & Wunsch 2006 showed how c. 100kyr periods can arise from multiples of the obliquity forcing, meaning that a signal of the same length as 'short' eccentricity doesn't necessarily mean it is driven by orbital eccentricity changes. For the caption, I would suggest to remove the speculation and just say '100 kyr cyclicity'.

Line 136-137: Related to above point – Fig A5 actually shows greatest power around 120 kyr period prior to 1 Ma, ie, most likely the third multiple of obliquity as per Huybers & Wunsch, and it is clearly quite distinct from the clear 100 kyr periodicity from 0.5 Ma onwards (which most likely IS eccentricity). [NB – eccentricity has its main components at 405, 95, and 124 kyr periods.]

Line 134-150: This entire paragraph first makes big assumptions about eccentricity being a dominant driver from 1.5 Ma, uses this assumption to make inferences about Thwaites connectivity, and then goes as far as to suggest that the AIS amplified eccentricity forcing prior to the MPT. Regardless of whether or not this is true, unfortunately the whole argument here is very speculative, because it relies simply on the assumption that any periodicity in the modelled ice volume signal that falls in the 80-120 kyr range is due to eccentricity, which certainly isn't beyond doubt. As mentioned in the general comments, I don't think any of the orbital arguments add anything to the paper, so I would cut this material out. If you do want to retain it for some reason, at the very least I would suggest reading Huybers & Wunsch and reframing this section in a way that more honestly acknowledges the uncertainty in ascribing oscillations of a particular frequency to a single causal driver.

Line 154: 'cascade' – see previous comments. I don't think any such phenomenon has been demonstrated, just say 'sequence of events'.

Line 170: please specify the value of the lapse rate, just so other modellers can be clear on the values used, rather than having to make an assumption.

Line 186-7: Could you explain why the mPWP anomaly is calculated with respect to the LIG, and what impact that might have on the results?

Line 219-20: I'm sure the authors are aware that LR04 doesn't record SL directly, so we need to know what is meant by 'derived from the benthic d18O stack' – how did you derive SL from the combined ice volume and deep ocean temperature record?

Figure A1 – all the figures in the paper are really beautiful, but I think this one would be a lot more convincing and easier to read if the panels were grouped by Type. At the moment they are colour-coded but I don't know what the colour groups correspond to?

Supplementary Info:

Line 6: can you explain what the likely impact of the thinner-than-observed modelled WAIS might be? I imagine it would make the response of all WAIS catchments much more sensitive to external forcings. This is pretty important, given that this is a key part of the paper. It would be good to have this aspect of uncertainty acknowledged in the main text.

Fig S1 – in addition to the above, it seems that the model is also simulating ice shelf flow that is too slow. What might the impact of this be on the response of the grounded ice? Again, acknowledging model limitations in the main text is important.

Fig S1 – why are there different ice extents in panels A and C?

Line 34: There is a figure call to S4 here, but S3 only comes later. Perhaps you might need to switch these figures around.

Fig S2 – it's really hard (almost impossible) to see the black grounding line over the darkest areas of panels A and B – can this be changed to something more visible (eg gold, or green, or white)?

Fig S4 caption – I think this figure needs much more explanation. It isn't clear to the reader what is core interpretation and what is modelled.

Fig S6 caption – ‘large differences in ocean temperature between the LIG and mPWP’ – how might this affect the results? If the early part of the record has very high variability in ocean temperatures, presumably this affects basal melt rates if/when grounded ice enters the ocean. Could this explain why several of the basins (pretty much all the marine basins of WAIS) don’t show increasing ice volume during this period (eg Fig 3, Fig A1)? This seems pretty important to me.

N. Golledge 5th March 2025

Reviewer #3

(Remarks to the Author)

“Cascade of abrupt transitions in Antarctic drainage basins before and during the Mid-Pleistocene Transition”

Author: Wirths, Herman, Stepanek, Stocker and Sutter

This study presents the role of the Antarctic ice sheet upon the Mid-Pleistocene Transition (MPT), which has not been explored much compared to that of Northern Hemisphere ice sheets. For the purpose, an Ice sheet model PISM, driven by a climate index method based on the snapshot results of COSMOS, is applied to perform 3 million-year simulations. Hysteresis in simulated volume for each drainage basin are discussed, and it is concluded that a tri-stability for Thwaites basin is a key factor to amplification of the earth orbital eccentricity during the MPT through the formation of marine ice sheets.

The focus is interesting, the discussion is mostly clear, thus it can be accepted with minor revision.

Major points

About ice-sheet model setup.

The experiment in the present paper seems to follow most of the configurations in Sutter et al. (2019) with various differences either written or not in the paper. In particular ice-sheet parameters such as enhancement factors are quite different between the two (see Tab.2 in Sutter and Tab. A1 in the paper). I wonder what is the meaning of the choices of the parameters in the paper. I suppose they come from the new tuning in terms of target-period coverage in the simulation (as noted in the Supplement), and/or the bore-hole temperatures (Fig. S3) which are not published at Sutter et al. (2019). Please clarify this situation for the particular setups in the current study.

Also the boundary conditions such as the reference climate or anomalies seem to be different. The present paper uses mPWP anomaly but the other uses Pliocene for the case when the climate index is greater than 1. LGM and LIG snapshots seem to be the same. I am not saying that the parameters must be identical, but wonder whether both studies are not contradictory. It might be possible that only the replacement of warmer conditions (mPWP and factor 1.3) leads to a completely different combination of the model setup, but probably not. Please clarify how and what affects the simulation with the combination.

Figure 3 (and also A1). I wonder how types are defined objectively, i.e., what ‘rapid’ means quantitatively. Vertical scales of Volume are all different in Figs 3, A1. Thus I suspect that the relative volumes are used to define the types.

Relating to above, The vertical axis of Figure A2 is confusing. Ice volumes are plotted in Figure 4, while ‘normalized’ volumes are plotted in A2. Moreover, this normalization seems to set the minimum volume as zero for each basin, which is not explained in the paper. Please clarify the point. Again, for example, I am confused that Basin 12 is type A while Basin 13 is type B, although their amplitudes (not normalized but absolute) are similar according to Figure A1.

Figure S7. As far as I understand, the region decomposition is for ocean forcing and may not relate to the one used for analysis, at least directly, in the paper (Basin 1-18). It is possible that the basin follows mostly the S7 with adjustment on Amery basin, etc. If so, please clarify this.

Figure 4.

Doesn’t it make sense to introduce a state density diagram such as Figure 3 also in Figure 4? It will be much clearer and objective to show the number of steady states for each basin.

Also, it is a little bit hard to compare Figures 3 and 4 to check the consistency of threshold levels., because horizontal/vertical axis and their ranges are not the same. Why don’t you include climate index values in either figure legend or subfigures b d f h?

If you agree to include the state density diagrams, it is probably better to move the surface topography subfigures to align horizontally under subfigure g, sorted by climate index (i.e., exchange d and f).

Eq. 1

These explanations are fine and correct, but not easy to understand at a glance.

I suggest append an interpretation text of the equation, and split in three regimes at value C_{llgm} to C_{lref} , C_{lref} to C_{llig} , and C_{llig} to C_{lmpwp} .

Equations (1) and (2) seem complex, however they merely tell that the temperature is linearly interpolated between T_{lgm} --

Tref, Tref -- Tlig, and Tlig -- Tmpwp for the three regimes above by climate index CI.
For $Cllgm < CI < Cllig$, T anomaly is weighted means of Tlgm and Tlig, which is added to the pre-industrial reference. On the other hand for $Cllig < CI$, T anomaly is weighted means of Tlig and Tmpwp (more precisely, $(CI - Cllig)/(CImpwp - Cllig)$), which is added to the LIG reference.

Minor points

L9, abstract. 800 ka BP should be 0.8 ka BP (which is used in the main text).

L57. 'Antarctic sea level equivalent ice volume was 8-9m lower...'.
I understand, but good to rephrase, because volume cannot be "lower".

Fig 2 legend. PISM should be something like 'this study', because you cannot be a representative of PISM among many PISM applications.

L76, (Thwaites, Ronne and Ross) should accompany the subfigure index e, f, g.

L 101. 'every 20 kyrs'. Is this duration sufficient to obtain steady-states? I am afraid that it may take longer, in particular in the climate near bifurcation points. Typical $dVolume/dt$ may be useful to check.

Figure 4 captions. Thin solid lines, thick solid lines.

Figure A3e. It is better to keep the ratio of actual R-R' to R'-R'' lengths in the vertical axis.

Figure A5. No explanation what the black dashed curves are (significant region?)

Supplement Reference. Lecavalier 2022 should be replaced with Lecavalier 2023 (ESSD, published).

Version 1:

Reviewer comments:

Reviewer #1

(Remarks to the Author)

The authors have responded appropriately to my comments, providing additional data and text.
I think this is an important paper for the field that deserves publication in Nature Communications.

Reviewer #2

(Remarks to the Author)

Wirths et al.

The revised manuscript has addressed all of the comments I made in the previous round of reviews, and I am grateful that the authors found the suggestions useful. The paper is now much less speculative (eg regarding the 'cascades' and the link to orbital components), and as such I think the results / discussion are much more robust. Overall it is a very comprehensive study and it will make a useful contribution to the literature on Antarctic ice sheet thresholds and long-term behaviour.

NG 12th Sept 2025

Reviewer #3

(Remarks to the Author)

The author team sufficiently revised the work and answered the questions and concern the reviewers raised. I am happy to recommend the work for publication in Nature Communications, as it meets the expected standards and it needs to be published in a hurry because of the expectation the community has.

Reply to comments by Reviewer 1: "Cascade of abrupt transitions in Antarctic drainage basins before and during the Mid-Pleistocene Transition"

Summary of Changes

We are grateful to the reviewer for evaluating our work, and the valuable and constructive comments that help improve the manuscript. To address the major comments, we now:

- Discuss the Ice thickness evolution since the Pliocene in light of exposure dating based reconstructions of past ice sheet elevation changes.
- Expand on the specifics of the stability of the Wilkes Subglacial Basin throughout our simulations.
- Provide a more detailed analysis of the grounding line dynamics within the Thwaites and Pine Island basin and further elaborate on the amplification of 100 kyr variability.

Below, we respond to the reviewer's individual comments in detail and describe the actions we took to address them.

Detailed response

(Original report cited in italics)

In this manuscript entitled, "Cascade of abrupt transitions in Antarctic drainage basins before and during the Mid-Pleistocene Transition" the authors aim to understand main drivers of Quaternary climate variability during the Mid-Pleistocene Transition (MPT) through the dynamics of the Pleistocene Antarctic ice sheet using regional Parallel Ice Sheet Models. The results indicate that the eccentricity cycle was particularly prominent in the Thwaites basin, West Antarctic Ice Sheet (WAIS) before the MPT, providing new insights into the distinctive behavior of the marine-based WAIS in response to climate fluctuation contributed to the MPT. This study improves the resolution of the history and dynamics of the Antarctic Ice Sheet and is expected to have a substantial impact on various research areas related to the global climate system.

Overall, this paper is well designed and the data are presented logically, but I have some questions and comments as described below.

Indeed, there are limited constraining data indicating Pleistocene Antarctic ice sheet fluctuations, but some geological reports exist (such as surface exposure ages). For example, observational data from DML suggest that around 3 million years ago, the inland region of East

Antarctica experienced an increase in ice sheet thickness. Subsequently, as the global cooling, the ice sheet elevation decreased, indicating a change in water transport and ice sheet morphology (Yamane et al., 2015, Nature Communications). How do you interpret the consistency between these findings and this paper?

As shown in Figure S2 (see below) of the Supplements, our simulations don't show any large-scale increase in East Antarctic ice thickness compared to the simulated present-day geometry. This is most probably due to an interplay of high Pliocene ocean temperatures which lead to a strong ice discharge through the drainage basins which offsets any surface mass balance increase. While increased Pliocene surface temperatures lead to higher accumulation rates (via the precipitation scaling), the warm ocean conditions dominate and lead to substantial ice loss in coastal regions (e.g. Weddel Sea, Amery shelf etc.) which propagates inland. One could explore this further by a larger ensemble investigating various precipitation scalings and ocean forcings but this goes beyond the scope of this study.

Figure S2: Mean ice thickness difference with respect to the end of simulation and observed present-day (gray) and simulated (black) grounding line position for (A) Pliocene (3Ma BP), (B) LIG (120 ka BP) and (C) LGM (19 ka BP).

In addition, a direct comparison of exposure age-based ice sheet elevation reconstructions and our simulations is complicated due to the grid-resolution (16 km) of our simulations. Nunataks are mostly local scale features which are not well resolved in the model, which leads to model biases at locations where exposure dating has been performed. For completeness, we have added the evolution of the ice sheet surface height at the EDML ice core location below and in the Supplements.

Figure S5: Surface elevation at the EDML ice core site throughout the last three million years. Thick line shows the ensemble mean, shaded lines individual ensemble members.

Line 59: From Fig. 2, it appears that this study presents lower values around 3 Ma to 2 Ma compared to de Boer et al. (2024). Is 'agreement' the appropriate term here?

The formulation might be confusing as the reference de Boer et al. (2015) is different from the simulations from de Boer et al. (2014) shown in Figure 1. Our statement emphasizes that simulated Antarctic Sea level contribution during the Pliocene is within the plausible range of Dutton et al. (2015) and the PLISMIP-ANT ensemble by de Boer et al. (2015). The spread between individual ice sheet models is in the order of several meters (de Boer et al., 2015), similar to the difference between our simulations and de Boer et al. (2014). To avoid confusion, we now write: “[...] in agreement with proposed sea level ranges from previous studies (de Boer et al., 2015; Dutton et al., 2015).”

• Lines 91–97: The absence of thresholds in the Wilkes Basin seems significant and provides various implications. While I understand that this may be due to topographic resolution (model resolution), it might be helpful to include additional figures illustrating these effects.

The drainage of ice in the Wilkes Subglacial Basin (WSB) is mainly routed through Ninnis, Cook and Mertz glaciers. Simulations of those outlet glaciers depend on the model resolution but also on the applied bedrock product. The impact of the spatial ice sheet model resolutions onto past and future stability has been examined in detail in Wang et al (2024) and Sutter et al. (2020). As Sutter et al. (2020) uses a similar model set-up, at higher resolution we especially refer to this publication here. The stability of the WSB is controlled by topographic pinning points and “ice plugs” (e.g. Mengel & Levermann, 2014) keeping the grounding line at the George V Coast in a stable configuration. Especially topographic pinning points, which are represented differently depending on the bedrock product, can affect the onset of Marine Ice Sheet Instability. Throughout our simulations, the George V coast grounding line remains slightly advanced during

interglacials and the ice sheet is stable (see below and Fig. S10 for LIG configuration), such that no large-scale collapse (MISI) of the Wilkes basin ice sheet is triggered. Overall, one could state that for the Wilkes Subglacial Basin our model setup is rather conservative and probably provides a lower bound with regard to interglacial sea level contributions from this region (see also Sutter et al., 2020, Crotti et al., 2022 and Lizuka et al., 2023).

We have added Fig. S10 and an explanatory text in the Supplementary Material to give a broader understanding of the WSB's model state during the Last Interglacial in our study.

Figure S10: Bedrock elevation and cross-sections through Wilkes subglacial basin and through the ice plug as in Mengel & Levermann (2014) during the Last Interglacial (120 ka BP) simulated under model configuration 1. The ice sheet configuration in both cross-sections reveals that the grounding line has not reached an unstable configuration.

- Ultimately, why does the Thwaites Basin ice sheet fluctuate with the eccentricity cycle? Additionally, does this mechanism differ from the variations in the Northern Hemisphere (Abe-Ouchi et al., 2013)? While I understand that complex interactions are involved, more detailed explanation be required. For instance, to what extent is the GIA component considered? What about dynamic elements? If the term 'dynamics' is used, the text include more discussion on factors such as flow velocity and ice sheet morphology.

The enhancement of 100 kyr variability within the Thwaites basin seems to originate from a combination of the unique climatic thresholds and the “overshooting stability” of the intermediate / proto-WAIS state (see Fig. 4). Under the “right” climatic conditions this can lead to skipping of interglacial collapses, enhancing variability at lower frequencies. As the climate index time series used in this study contains variability at various periodicities, we have performed an additional simulation with a synthetic climate index using the long-term trend of the climate index time series with an artificial 41 kyr sinusoidal signal. We added the synthetic climate index and the resulting ice volume evolution in the Thwaites basin for model configuration No. 1 to supplementary Fig. S12 (and also show it below). We infer that for the synthetic climate index, oscillating between interglacial values of 0.60-0.67 and glacial values of 0.20-0.27, skipping of interglacial ice sheet collapse can be observed, although the occurrence of “skipping” does not seem to be of deterministic nature. We attribute the latter to the complexity of the three-dimensional ice flow within the basin.

We have now expanded the Supplementary Material by Figure S12 and added an additional paragraph describing the process.

Figure S12: Tracing enhancement of 100 kyr variability within the Thwaites basin. We show a synthetic climate Index created by adding a 41 kyr sinusoidal onto the long-term trend of the original climate index (top) and the resulting ice volume within the Thwaites basin (bottom). For climate index values between 0.20-0.27 (glacial) and 0.60-0.67 (interglacial), the unique thresholds in the basin and their temporal stability can lead to skipping of interglacial collapses, ultimately leading to enhanced lower-frequency (100 kyr) variability.

To assess the origin of the unique thresholds within the Thwaites and Pine Island glacier we have illustrated the ice-bedrock configurations along two cross-sections through the respective glaciers. Both the simulated present-day configuration and the LGM and Proto-WAIS / intermediate states are shown in Figure S11, see also below. As becomes apparent, in all

configurations the ice sheet seems to rest on a prograde slope, resulting in a stabilized grounding line position. Especially, the intermediate state of Thwaites glacier is in a partly advanced configuration on this prograde slope, which might explain its enhanced stability under paleo-overshooting on glacial-interglacial time scales before and during the MPT. In this regard the results may be affected by the choice of Earth deformation model. In this study, we have utilized the Lingle-Clark model (Lingle & Clark, 1985; Bueler et al., 2007) which accounts for the bedrock response to changing ice loads with a single parameter for viscosity that can only partly reflect regionally heterogenous bedrock strengths as derived e.g. from 3D GIA modelling (see e.g. Gomez et al. 2024, Science Advances). To address the discussed topics above we now have added Figure S11 together with a paragraph to the supplementary material.

We sincerely thank the reviewer for their thoughtful and constructive comments, which we believe have substantially enhanced the overall clarity and quality of the manuscript.

Figure S11: Bedrock elevation and cross-sections through Thwaites (Sec A) and Pine Island (Sec B) subglacial basin for simulated present day ice sheet (grey) and bedrock (brown) configurations, as well as the ice sheet configuration during the LGM (green) and the Proto WAIS state (red) simulated under model configuration 1. In all cases the grounding line stabilizes at the onset of a prograde bedrock slope that limits the advancement of the grounding line. The grounding line in the Proto WAIS configuration (Sec A) seems to be also positioned on a prograde bedrock slope that might support the increased stability during past “overshooting” events of that specific basin.

Reply to comments by Reviewer 2: "Cascade of abrupt transitions in Antarctic drainage basins before and during the Mid-Pleistocene Transition"

Summary of Changes

We are grateful to the reviewer for evaluating our work, and for providing valuable and constructive comments that help improve the manuscript. To address the major comments, we now:

- provide a more precise explanation of our classification scheme for individual basin types and clarify the intention behind this categorization.
- replaced the term "Cascade" consistently with "Sequence of events" throughout the manuscript, including the title.
- specified the interpretation of the spectral analysis, carefully avoiding direct attribution of specific frequencies to orbital parameters.

Below, we respond to the reviewer's individual comments in detail and describe the actions we took to address them.

Detailed response

(Original report cited in italics)

The paper presents a suite of long (3 My) transiently-forced ice sheet model experiments that simulate the evolution of the Antarctic Ice Sheet through the end-Pliocene and whole of the Pleistocene. From these model runs the authors extract ice volume timeseries for each discrete

ice sheet drainage basin, and use the patterns in these timeseries to infer distinct ‘types’ of catchment. By reversing their experiments, the authors also explore hysteresis, and by using spectral analysis they make connections between modelled transient behaviour and Milankovitch cyclicities.

The paper is generally well-written and the figures are of a very high quality. The simulations are impressive – doing long transient simulations such as these is not an easy undertaking. And the ideas presented are interesting and certainly add something to the discussion around the nature of the Mid-Pleistocene Transition and how the AIS responded at this time. Hysteresis experiments (long, whole continent) and catchment-scale AIS modelling (shorter, regional scale) have been done for several decades now, and there are a lot of papers on these topics (although not many are referenced, see below). The advance here I think is doing both these things together – very long simulations analysed at the catchment scale. So it is a valuable contribution in that sense. I do have some reservations about the way that some of the arguments are constructed however, and outline these below.

General comments

1) To me it seems that there are three parts to this paper: i) the identification of catchment-specific behaviours, ii) modelling hysteresis, and iii) orbital forcing. Of these, I found the first to be most compelling. I like the basin-level insights. But it isn’t made clear what the hysteresis experiments actually add to the story, other than showing that the same series of events plays out in reverse. The orbital inferences I think are very weak, and this aspect either needs to be much more rigorously and thoroughly explored, or left out completely (see detailed comments below).

Please find the response to the individual points below.

2) Evidence for cascades: to me, there is no evidence presented anywhere in the paper that supports the idea of a ‘cascade’ (in the sense that events in one basin trigger events in another, that then trigger another and so on). I expand on this below, but given this fundamental issue, I think the title and emphasis of the paper needs to be addressed. What is being shown is a sequential response of regionally-abstracted metrics to the imposed continent-wide forcing.

We agree that based on this definition of ‘cascade’ our title could be misinterpreted. Our intended idea of the word ‘cascade’ was rather aimed at longer time periods: throughout the simulations, more and more thresholds are crossed within the individual Antarctic basins. To avoid a misinterpretation of the word ‘cascade’ we changed the title of the manuscript to: “Sequence of abrupt transitions [...]”.

3) Importance of model setup: like most simulations that employ a spinup rather than data assimilation, the model used here doesn’t agree perfectly with observations – it is too thin in WAIS and the ice shelves flow too slowly. That’s totally fine, but I think it is important to have some discussion in the main text on how this affects the results. Similarly, see the final comment below on 3-2 Ma ocean temperature variability in the applied forcing – the peaks are very warm and most likely prohibit GL advance for WAIS. The implications of this need to be explained in the main text.

We now discuss potential biases and influences of modeling choices and uncertainties within the manuscript. See line 161 and following of the revised manuscript.

4) The referencing is ok, but misses a lot of other work that is relevant. For example, I counted only about a dozen of the > 50 references in the main paper that are AIS modelling studies – the rest are mostly climate modelling, proxy, or theoretical / methodological studies. Given that the focus of the manuscript is on modelling the AIS, and that some novelty is required to justify publication, I would suggest that a more in-depth reading of the literature might be useful to show a) what else has been done previously regarding AIS catchment types and catchment-scale AIS tipping points / transitions, and b) what the new study adds to that.

We now refer to, and discuss, additional ice sheet modeling studies that investigated the dynamics of the AIS, as a whole or of individual sub basins, and associated thresholds within time periods since the Pliocene (e.g. Golledge et al., 2014; Mengel & Levermann, 2014; Sutter et al., 2016; Golledge et al., 2017; Lowry et al., 2019; Golledge et al., 2021).

5) There are quite a few typos and grammatical errors (especially in the SI material) – I haven’t detailed them below and so I would suggest a careful line-edit to find and address these.

We carefully checked the whole manuscript and supplement and corrected typos and grammatical errors. Furthermore, we improved the overall style of the text.

Specific comments

Title: Might need to revisit this in light of comments below about lack of evidence for ‘cascades’

We have changed any references to the term cascade to the term sequence.

Line 12-13: ‘eccentricity-paced’ – I have some issues with this, see later comments.

In alignment with the comment below, we don’t use the term “eccentricity paced” anymore.

Line 17: In addition to Berends, perhaps it would be worth reading the more recent and comprehensive paper by Herbert 2023.

We now additionally cite Herbert et al. (2023).

Fig. 1 and elsewhere. Might be worth mentioning that Ahn et al 2017 effectively supersedes LR04

In this manuscript, we consistently used the data from Lisiecki and Raymo (2005) and therefore refer only to this data set. While Ahn et al. (2017) provide an updated data set, we consider it very similar to LR04 and do not see a significant benefit in mentioning an alternative data set not applied within this study.

Line 61: I think the onset of the Pleistocene is closer to 2.6 Ma (2.58 Ma according to the International Strat. Commission).

Corrected.

Line 74: I think ‘phase space density’ needs to be introduced and explained, because it is not an intuitive metric, and a lot of the discussion hangs on this.

We added a description of the phase space density metric to that section. Line 79 now reads: “The phase space density quantifies the frequency of occurrence of a specific combination of ice volume and climate index through time, considering all six simulations.”

Line 78: How exactly are these types identified? Is it simply a visual grouping based on perceived similarity? Or is it quantified in some way? I think it needs to be defined and justified. The problem I think is that grouping based on the timeseries appearance is very subjective, and grouping by state density histogram shape could yield different results. For example, in Fig. 3, Thwaites histogram looks more like a Type B than a Type C, whilst Ross (Type B) actually also looks quite like Ronne (Type C). Perhaps some objective approach can be found to distinguish between the catchments?

Our classification of the observed basin response was intended to highlight similarities and differences between individual basins. As such a classification is not intended to, and indeed does not represent any physical law, the definition of those types will, to a degree, always be subjective. To classify the individual basins, we utilize the ice volume time series from Fig. 3 and Fig. A1 and later verify the crossing of thresholds in our hysteresis experiments.

We categorize basins, that show a slow gradual change in ice sheet variability throughout time due to the absence of crossing a major threshold, as basins of Type A.

Basins of Type B are characterized by the abrupt onset of variability due to the crossing of a threshold at a certain climate index level.

Basins of Type C show several abrupt increases of variability, that stem from crossing of thresholds, and an additional intermediate meta stable state.

We don't utilize the histograms for that classification as the unequally distributed climate index might exaggerate or dampen the occurrence density of certain states. To avoid confusion, we now emphasize in L. 83 that we rely on the ice volume time series when performing this classification, instead of just broadly referring to our results.

Line 84: 'three preferred states' – this isn't clear for Ronne, and the histogram really only shows two. Again, I would suggest finding a way to formalise the way these states are defined.

For the Ronne basin, the glacial state seems not to be “preferred” as it doesn't occur in the histogram, which is related to the relatively rare occurrence of climatic conditions that cause the basin to reach this state and stay in such a configuration. However, both the ice volume time series and the hysteresis fulfill the above-mentioned criteria of a Type C basin. We therefore omit the label “preferred” in the revised version of the manuscript.

Line 121-123: This is really the only place in the manuscript which talks about the so-called 'cascade', but to me it is simply a sequential pattern of change – there is no mechanistic or causal link demonstrated. The problem I think is that both sectors (Ross and Thwaites) are being forced synchronously with the same timeseries perturbation, so it is natural that they will both respond to that forcing in a way that is conditioned by their geometry & dynamic configuration. To show a causal link, you'd probably need to have an experiment where the forcing was only applied in one of these sectors, to see whether changes in A lead to changes in B, which trigger further changes in A etc. It might be easier to just avoid saying there is a cascade...

We agree that a more elaborate framework or analysis would be needed to justify the word cascades and changed the wording accordingly (see response to 2)).

Line 126-131: It seems that this paragraph is the only text to explore and justify the hysteresis experiments, but I still don't really understand what these 3 sentences in particular are saying: "Notably, the intermediate ice volume state exhibits a high phase space density during our transient simulations (Fig. 4 c). This phenomenon can only be partially explained by a hysteresis effect, as demonstrated by our reversal experiments on the intermediate state (see. Fig. A4). These findings suggest that the intermediate state of Thwaites basin can be resilient against overshooting its equilibrium state threshold." Can this be clarified, or just removed?

Our reasoning is as follows: We perform hysteresis experiments to determine whether each AIS basin's response to changing climate over the last 3 million years simply reflects equilibrium with the climate, or if differences in the timing of climate changes, and of the ice sheet's response, could have caused additional dynamics. These experiments also help to better understand and locate thresholds that trigger substantial reorganizations within individual basins and help explain the differences in the ice volume time series shown in Fig. 3.

To clarify the motivation for the hysteresis experiment we have expanded the text in Line 100 onwards, which now reads: "To validate and quantify the thresholds that trigger substantial reorganizations within individual AIS basins and associated reversibility, as well as to investigate whether differences in the timing of climate changes and ice sheet responses could have led to additional dynamics beyond simple equilibrium behavior, we performed a series of hysteresis experiments. Specifically, we conducted a stepwise quasi-equilibrium hysteresis simulation, incrementally adjusting the climatic forcing index by 0.05 every 20 kyrs in both directions (cooling, warming), for all six model configurations."

Fig. 5 – Are there units for the colour bar? I'm not sure what these 'standardized' numbers mean.

To facilitate easier comparison of the spectral power between the individual basins and the climate index, prior to the spectral analysis all time-series were standardized towards a zero mean value and unit variance. As a result, the spectral power is expressed in units of inverse frequency, corresponding to time. We have updated the figure accordingly. To avoid any confusion, we now explicitly mention the standardization procedure in the caption of Fig. 5.

Fig 5 caption – it is fine to state the window of the different spectral bands (eg 80-120, 39-43), but it is not clear that they can be unequivocally linked to eccentricity and obliquity respectively. Huybers & Wunsch 2006 showed how c. 100kyr periods can arise from multiples of the obliquity forcing, meaning that a signal of the same length as 'short' eccentricity doesn't necessarily mean it is driven by orbital eccentricity changes. For the caption, I would suggest to remove the speculation and just say '100 kyr cyclicity'.

We agree that directly linking the variability bands to obliquity and eccentricity can be misleading. Therefore, we now just refer to ~100 and ~40 kyr variability/cyclicity within our manuscript.

Line 136-137: Related to above point – Fig A5 actually shows greatest power around 120 kyr period prior to 1 Ma, ie, most likely the third multiple of obliquity as per Huybers & Wunsch, and it is clearly quite distinct from the clear 100 kyr periodicity from 0.5 Ma onwards (which most likely IS eccentricity). [NB – eccentricity has its main components at 405, 95, and 124 kyr periods.]

Please refer to our answer above.

Line 134-150: This entire paragraph first makes big assumptions about eccentricity being a dominant driver from 1.5 Ma, uses this assumption to make inferences about Thwaites connectivity, and then goes as far as to suggest that the AIS amplified eccentricity forcing prior to the MPT. Regardless of whether or not this is true, unfortunately the whole argument here is very speculative, because it relies simply on the assumption that any periodicity in the modelled ice volume signal that falls in the 80-120 kyr range is due to eccentricity, which certainly isn't beyond doubt. As mentioned in the general comments, I don't think any of the orbital arguments add anything to the paper, so I would cut this material out. If you do want to retain it for some reason, at the very least I would suggest reading Huybers & Wunsch and reframing this section in a way that more honestly acknowledges the uncertainty in ascribing oscillations of a particular frequency to a single causal driver.

We agree with the reviewer. In alignment with our replies above, we now don't speculate about specific orbital parameters being the origin for certain changes in variability that we observe. We now only refer to individual variability bands without linking them to any orbital origin.

Line 154: 'cascade' – see previous comments. I don't think any such phenomenon has been demonstrated, just say 'sequence of events'.

The title now reads "Sequence of events [...]"

Line 170: please specify the value of the lapse rate, just so other modellers can be clear on the values used, rather than having to make an assumption.

We added the value for the dry adiabatic lapse rate of 9.8 K/km to the sentence.

Line 186-7: Could you explain why the mPWP anomaly is calculated with respect to the LIG, and what impact that might have on the results?

The climate index method uses the anomaly between the mPWP and LIG to achieve a smooth transition in the climatic forcing between the LIG and mPWP. As the climate index applies the spatial pattern of the anomalies, this way there is a gradual progression from the LIG anomaly pattern to the mPWP anomaly pattern. Had we used other definitions then this would pose the potential for a discontinuous break in the spatial distribution of the anomalies. For any index value larger than 1, the weight ω_{LIG} is always 1, thus adding the LIG anomaly to the reference forcing. We clarified this shortly within the revised manuscript which now reads: “[...] for the mPWP the anomaly is calculated with respect to the LIG to produce a smooth and seamless transition.”..

Line 219-20: I'm sure the authors are aware that LR04 doesn't record SL directly, so we need to know what is meant by 'derived from the benthic $\delta^{18}O$ stack' – how did you derive SL from the combined ice volume and deep ocean temperature record?

We agree with the reviewer that the benthic $\delta^{18}O$ stack reflects a combined signal of ice volume and deep-ocean temperature, and of not sea level directly. To derive a sea-level forcing, we used a linear scaling approach inspired by work in Forster and Rohling (2013), calibrated to produce sea level anomalies wrt. present-day of approximately –115 m at the Last Glacial Maximum and +8 m at the Last Interglacial. These values align with independent reconstructions and studies from Foster et al. (2009) and Dutton et al. (2015). While the assumption of linearity is a simplification, it would be difficult to derive a transient scaling on the timescales considered here. E.g. A scaling tailored for the Pliocene is possible but, given the substantial uncertainties in Pliocene sea-level reconstructions, it would not necessarily improve the reliability of our forcing. To clarify all these details in our manuscript we now expanded the description of the sea level forcing within the Methods section of the main manuscript and within the Supplementary Material.

Figure A1 – all the figures in the paper are really beautiful, but I think this one would be a lot more convincing and easier to read if the panels were grouped by Type. At the moment they are colour-coded but I don't know what the colour groups correspond to?

The color-coding is a remaining artefact of an earlier version of the figure and was kept for aesthetical reasons and to make it easier for the reader as a specific basin is always associated with a corresponding color (e.g. Thwaites in brown, Ross in purple etc.).

Following the suggestion by the reviewer we have now regrouped the basins by the assigned Type in Fig. A1, Fig. A2 and Fig. 3, respectively.

Supplementary Info:

Line 6: can you explain what the likely impact of the thinner-than-observed modelled WAIS might be? I imagine it would make the response of all WAIS catchments much more sensitive to external forcings. This is pretty important, given that this is a key part of the paper. It would be good to have this aspect of uncertainty acknowledged in the main text.

Please refer to our reply below.

Fig S1 – in addition to the above, it seems that the model is also simulating ice shelf flow that is too slow. What might the impact of this be on the response of the grounded ice? Again, acknowledging model limitations in the main text is important.

The lower than observed ice shelf velocities indeed could bias the stability of the grounding line and the response of the individual basins towards changing climatic conditions. We now discuss this in the main text (Line 161 onwards).

Fig S1 – why are there different ice extents in panels A and C?

We now show anomaly aligned with the observed (BedMachine, Morlighem et al., 2020) observations.

Line 34: There is a figure call to S4 here, but S3 only comes later. Perhaps you might need to switch these figures around.

Thanks for pointing us to this, figures have been rearranged accordingly.

Fig S2 – it's really hard (almost impossible) to see the black grounding line over the darkest areas of panels A and B – can this be changed to something more visible (eg gold, or green, or white)?

We adapted the colors for enhanced visibility.

Fig S4 caption – I think this figure needs much more explanation. It isn't clear to the reader what is core interpretation and what is modelled.

Fig. S4 (now S3) shows simulated data at the AND-1B site. We now emphasise this fact, in addition to the caption, also in the accompanying text in the supplements. There, our results are discussed in the context of the AND-1B sediment core (Naish et al., 2009; McKay et al., 2012).

Fig S6 caption – 'large differences in ocean temperature between the LIG and mPWP' – how might this affect the results? If the early part of the record has very high variability in ocean temperatures, presumably this affects basal melt rates if/when grounded ice enters the ocean. Could this explain why several of the basins (pretty much all the marine basins of WAIS) don't

show increasing ice volume during this period (eg Fig 3, Fig A1)? This seems pretty important to me.

The high ocean temperature anomalies for climate index values in between the LIG and mPWP reference value, definitely lead to a retreat of the grounding line in many Antarctic basins and especially to a collapse of the WAIS right at the onset of the model simulations. However, for many basins (e.g. WAIS) we observe a collapse already for the relatively colder LIG conditions. Therefore, we assume the impact of the “warmer than LIG” mPWP ocean conditions on the AIS growth during the Pliocene to be only of smaller importance. For example, for the WAIS, Fig. 4 clearly shows that, to advance the ice sheet from a collapsed state, the climate index value needs to fall below Last Interglacial conditions. It would be interesting to explore the specific melt rate thresholds (especially given the various sensitivities of different melt rate parameterizations) in a more expansive ensemble that is beyond the scope of this study.

N. Golledge 5th March 2025

We sincerely thank the reviewer for their insightful comments and valuable feedback. The suggestions have significantly improved the clarity and quality of the manuscript. We greatly appreciate the time and effort invested by the reviewer in evaluating our work.

Reply to comments by Reviewer 3: "Cascade of abrupt transitions in Antarctic drainage basins before and during the Mid-Pleistocene Transition"

Summary of Changes

We are grateful to the reviewer for evaluating our work, and the valuable and constructive comments that help improve the manuscript. Below, we respond to the reviewer's individual comments in detail and describe the actions we took to address them.

Detailed response

(Original report cited in italics)

"Cascade of abrupt transitions in Antarctic drainage basins before and during the Mid-Pleistocene Transition"

Author: Wirths, Herman, Stepanek, Stocker and Sutter

This study presents the role of the Antarctic ice sheet upon the Mid-Pleistocene Transition (MPT), which has not been explored much compared to that of Northern Hemisphere ice sheets. For the purpose, an Ice sheet model PISM, driven by a climate index method based on the snapshot results of COSMOS, is applied to perform 3 million-year simulations.

Hysteresis in simulated volume for each drainage basin are discussed, and it is concluded that a tri-stability for Thwaites basin is a key factor to amplification of the earth orbital eccentricity during the MPT through the formation of marine ice sheets.

The focus is interesting, the discussion is mostly clear, thus it can be accepted with minor revision.

Major points

About ice-sheet model setup.

The experiment in the present paper seems to follow most of the configurations in Sutter et al. (2019) with various differences either written or not in the paper. In particular ice-sheet parameters such as enhancement factors are quite different between the two (see Tab.2 in Sutter and Tab. A1 in the paper). I wonder what is the meaning of the choices of the parameters in the paper. I suppose they come from the new tuning in terms of target-period coverage in the simulation (as noted in the Supplement), and/or the bore-hole temperatures (Fig. S3) which are not published at Sutter et al. (2019). Please clarify this situation for the particular setups in the current study.

Indeed, there are various differences between this study and the earlier work by Sutter et al. (2019) and Sutter et al. (2020). In fact, also the work by Wirths et al. (2024) provides rationale for our setup choice. In the current manuscript, as in Sutter et al (2019) and Sutter et al. (2020), we use the climate-index method to drive the ice sheet model with simulated climate forcing across a time period of interest. While Sutter et al. (2019) focused on long-term ice sheet evolution across the mid-Pleistocene transition and towards today, Sutter et al. (2020) studied Antarctic Ice Sheet evolution during the Last Interglacial. An important difference between their studies and ours is the ice sheet model version, where we employ the more recent PISM 2.1, while the simulations in Sutter et al. (2019) were performed using the now outdated PISM 0.73. As a further difference to the method by Sutter et al. (2019) we now employ updated mid-Pliocene climate model output that was not yet available at the time. The new climate model output by Stepanek et al. (2020) reflects updates in the Pliocene geography employed in the Model (Dowsett et al., 2016) and now includes climate-vegetation feedbacks via employment of a dynamic vegetation scheme (that was absent in the earlier simulation due to choices made by the PlioMIP1 protocol; Haywood et al., 2010; Haywood et al., 2011). As a result, climate forcing imposed on the ice sheet model does reflect temperature dynamics at high latitudes more realistically, and there is more consistency between climate forcings of Pliocene and of other periods involved in the construction of the climate index, where in climate simulations vegetation dynamics were present as well.

While for most ice sheet model parameters we applied the (recommended) default values of PISM 2.1, we choose specific ice flow parameters (sia_e, ssa_e, pQ) and a bed roughness

parameter (ϕ_{\min}) to fit the simulated ice sheet to present-day observations (e.g. grounding line position, ice thickness, total ice volume, rate of ice volume change) as in Wirths et al. (2024). The reviewer is right that so far we only mention these details in the Methods Section and not in Tab. A1. For improved clarity we now also present this information in Tab. A1 in the revised manuscript.

The borehole temperature profiles were not used to tune the ice sheet model but indicate a robust fit of the model compared to the available direct observations at the corresponding deep drill sites

Also the boundary conditions such as the reference climate or anomalies seem to be different. The present paper uses mPWP anomaly but the other uses Pliocene for the case when the climate index is greater than 1. LGM and LIG snapshots seem to be the same. I am not saying that the parameters must be identical, but wonder whether both studies are not contradictory. It might be possible that only the replacement of warmer conditions (mPWP and factor 1.3) leads to a completely different combination of the model setup, but probably not. Please clarify how and what affects the simulation with the combination.

The climate-index method described by Sutter et al. (2019) allows the usage of “a warmer than interglacial climate” (e.g. Pliocene), just like in this study however it plays a smaller role in their study mainly because their simulations start at 2Ma before present where the climate index is significantly lower than at the mPWP. Additionally, in Sutter et al., 2019 an older GCM output for the mPWP is employed (see comment above). The reference values are just scaling values towards constructing the index. We associate the climatic conditions of the mPWP with the climate index value of 1.2 (see Fig 2.) and then consistently use the same reference value with the mPWP snapshot to calculate the climatic forcing. Thus, we don't contradict the climate-index methodology from Sutter et al. (2019).

Figure 3 (and also A1). I wonder how types are defined objectively, i.e., what 'rapid' means quantitatively. Vertical scales of Volume are all different in Figs 3, A1. Thus I suspect that the relative volumes are used to define the types.

We have refined the descriptions and definitions of the individual types, see also our response to N. Golledge (Reviewer 2) and updated the main text of the manuscript (see Line 83 and following). In this context, we identify climatic thresholds that mark the onset of substantially increased ice-volume variability and, depending on the basin, shifts in mean ice volume on glacial–interglacial timescales. These findings indicate that the basin has transitioned into a different dynamical regime or state. As stated in our reply to N. Golledge (Reviewer 2), our classification into Types is intended to provide readers with a synthesized overview of dynamical

similarities and differences between individual basins, rather than to represent a mechanistic process description. When using the term “rapid”, we do not refer to a specific velocity or rate of change but rather to the character of the transition, which becomes pronounced (akin to a step-function) and comparatively swift once a climatic threshold is crossed.

Relating to above, The vertical axis of Figure A2 is confusing. Ice volumes are plotted in Figure 4, while `normalized` volumes are plotted in A2. Moreover, this normalization seems to set the minimum volume as zero for each basin, which is not explained in the paper. Please clarify the point. Again, for example, I am confused that Basin 12 is type A while Basin 13 is type B, although their amplitudes (not normalized but absolute) are similar according to Figure A1.

The ice volume in Figure A2 is “Min-Max” scaled to allow the illustration of the hysteresis in all basins of different total ice volume on one common scale. We now clarify this in the figure caption.

Basin 12 is indeed a Type B per our classification as there is a substantial increase in variability and a change in the long term mean state at around 2.5 Ma BP. We have corrected Fig. A1 and A2 accordingly.

Figure S7. As far as I understand, the region decomposition is for ocean forcing and may not relate to the one used for analysis, at least directly, in the paper (Basin 1-18). It is possible that the basin follows mostly the S7 with adjustment on Amery basin, etc. If so, please clarify this.

Indeed, the map in Fig. S7 shows a modification from the IMBIE (Rignot et al., 2011) basins used to create the oceanic forcing. In this version several basins (14 & 15, 6 & 11, 8 & 13, 1 & 9) have been merged to ensure that all basins are connected to the open ocean.

Figure 4.

Doesn't it make sense to introduce a state density diagram such as Figure 3 also in Figure 4? It will be much clearer and objective to show the number of steady states for each basin.

In fact, Fig. 4 contains the phase space density (color shading) from Fig. 3 to allow a comparison between the dynamic/transient response of the ice sheet to glacial interglacial scale variations with an (quasi-)equilibrium response.

Drawing an additional phase space density from the hysteresis alone wouldn't in our opinion bring any considerable benefit, as contrary to the transient simulation each climate index value is only visited twice (cooling and warming branch). In our opinion, this would make a comparison of the transient simulation with the (quasi-)equilibrium response more complicated. However, we agree that in the current manuscript this notion is not clearly communicated. We therefore adjusted the paragraph as stated in our reply to N. Golledge (Reviewer 2). Line 100 onwards now

reads: “To validate and quantify the thresholds that trigger substantial reorganizations within individual AIS basins and associated reversibility, as well as to investigate whether differences in the timing of climate changes and ice sheet responses could have led to additional dynamics beyond simple equilibrium behavior, we performed a series of hysteresis experiments. [...]”

Also, it is a little bit hard to compare Figures 3 and 4 to check the consistency of threshold levels., because horizontal/vertical axis and their ranges are not the same. Why don't you include climate index values in either figure legend or subfigures b d f h?

Thank you for that suggestion. We know label the ice sheet configuration plots (Panel b,d,f,h) with the corresponding climate index values.

If you agree to include the state density diagrams, it is probably better to move the surface topography subfigures to align horizontally under subfigure g, sorted by climate index (i.e., exchange d and f).

Eq. 1

These explanations are fine and correct, but not easy to understand at a glance.

I suggest append an interpretation text of the equation, and split in three regimes at value CI_{lgm} to CI_{ref} , CI_{ref} to CI_{lig} , and CI_{lig} to CI_{mpwp} .

Equations (1) and (2) seem complex, however they merely tell that the temperature is linearly interpolated between T_{lgm} -- T_{ref} , T_{ref} -- T_{lig} , and T_{lig} -- T_{mpwp} for the three regimes above by climate index CI .

For $CI_{lgm} < CI < CI_{lig}$, T anomaly is weighted means of T_{lgm} and T_{lig} , which is added to the pre-industrial reference. On the other hand for $CI_{lig} < CI$, T anomaly is weighted means of T_{lig} and T_{mpwp} (more precisely, $(CI - CI_{lig}) / (CI_{mpwp} - CI_{lig})$), which is added to the LIG reference.

Thank you for highlighting the benefit of additional clarifications. To summarize what Eq. 1 and Eq. 2 state we have now added the following text after Eq. 2 in order to descriptively summarizing them: “In summary, this procedure generates a climate forcing field for a given climate index by interpolating between the two adjacent climate model snapshots.”

Minor points

L9, abstract. 800 ka BP should be 0.8 ka BP (which is used in the main text).

Now reads as “[...] 0.8 Ma BP, [...]”.

L57. `Antarctic sea level equivalent ice volume was 8-9m lower...`.

I understand, but good to rephrase, because volume cannot be “lower”.

We now write “smaller”.

Fig 2 legend. PISM should be something like `this study`, because you cannot be a representative of PISM among many PISM applications.

Adapted.

L76, (Thwaites, Ronne and Ross) should accompany the subfigure index e, f, g.

Added.

L 101. `every 20 kyrs`. Is this duration sufficient to obtain steady-states? I am afraid that it may take longer, in particular in the climate near bifurcation points. Typical $dVolume/dt$ may be useful to check.

The duration of 20 kyr refers to the relaxation time after each small incremental step of 0.05 in the climate index. We agree that, in principle, especially near bifurcation points, longer equilibration times would be desirable, since in theory a tipping-point crossing could occur only after 100 kyr, or even 1 Myr, of constant forcing. However, such timescales are computationally not feasible in the context of our study, as completing a full hysteresis cycle already requires approximately 1 Myr of simulation time.

Importantly, the pacing of our hysteresis experiments is comparable to, and even slower than, similar studies. For instance, Garbe et al. (2020) applied a GMT change rate of 0.0001 °C/yr. Translated to a regional oceanic temperature increase of about 5 °C (the glacial–peak-interglacial range in our simulations; see Fig. S7), this corresponds to ~70 kyr (please note that oceanic and global temperatures are not scaled one to one) of warming. By contrast, our approach requires ~400 kyr for the same temperature change.

Furthermore, our hysteresis pacing (a full cycle lasting ~ 1 Myr) is substantially slower than the rates of change in our transient glacial–interglacial simulations. We are therefore confident that our setup is sufficiently slow to capture all relevant transitions in Antarctic Ice Sheet dynamics under the considered climate forcing scenarios.

Figure 4 captions. Thin solid lines, thick solid lines.

Added.

Figure A3e. It is better to keep the ratio of actual $R-R'$ to $R'-R''$ lengths in the vertical axis.

Adapted accordingly.

Figure A5. No explanation what the black dashed curves are (significant region?)

Shaded regions and black dashed lines outline the cone of influence for which the wavelets analysis can be performed reliably. We have added this information to the figure caption.

Supplement Reference. Lecavalier 2022 should be replaced with Lecavalier 2023 (ESSD, published).

Adapted.

We thank the reviewer for their detailed and constructive comments. We are confident that their feedback has helped improve the manuscript. We appreciate their time and effort in reviewing our work.

References:

Bueler, E., Lingle, C. S. & Kallen-Brown, J. A. Fast computation of a viscoelastic deformable Earth model for ice sheet simulation. Ann. Glaciol. 46, 97–105 (2007).

de Boer, B., Dolan, A. M., Bernales, J., Gasson, E., Goelzer, H., Golledge, N. R., Sutter, J., Huybrechts, P., Lohmann, G., Rogozhina, I., Abe-Ouchi, A., Saito, F. & van de Wal, R. S. W. Simulating the Antarctic ice sheet in the late-Pliocene warm period: PLISMIP-ANT, an ice-sheet model intercomparison project. The Cryosphere 9, 881–903 (2015).

de Boer, B., Lourens, L. J. & van de Wal, R. S. W. Persistent 400,000-year variability of Antarctic ice volume and the carbon cycle is revealed throughout the Plio-Pleistocene. Nat. Commun. 5, 2999 (2014).

Crotti, I., Quiquet, A., Landais, A. et al. Wilkes subglacial basin ice sheet response to Southern Ocean warming during late Pleistocene interglacials. Nat. Commun. 13, 5328 (2022).

Dowsett, H., Dolan, A., Rowley, D., Moucha, R., Forte, A. M., Mitrovica, J. X., Pound, M., Salzmann, U., Robinson, M., Chandler, M., Foley, K., and Haywood, A. The PRISM4 (mid-Piacenzian) paleoenvironmental reconstruction. Clim. Past, 12, 1519–1538 (2016).

Dutton, A., Carlson, A. E., Long, A. J., Milne, G. A., Clark, P. U., DeConto, R., Horton, B. P., Rahmstorf, S. & Raymo, M. E. Sea-level rise due to polar ice-sheet mass loss during past warm periods. *Science* 349, aaa4019 (2015).

Foster, G. L. & Rohling, E. J. Relationship between sea level and climate forcing by CO₂ on geological timescales. *Proc. Natl Acad. Sci. USA* 110, 1209–1214 (2013).

Garbe, J., Albrecht, T. & Levermann, A. et al. The hysteresis of the Antarctic Ice Sheet. *Nature* 585, 538–544 (2020).

Haywood, A. M., Dowsett, H. J., Otto-Bliesner, B., Chandler, M. A., Dolan, A. M., Hill, D. J., Lunt, D. J., Robinson, M. M., Rosenbloom, N., Salzmann, U., and Sohl, L. E. Pliocene Model Intercomparison Project (PlioMIP): experimental design and boundary conditions (Experiment 1). *Geosci. Model Dev.*, 3, 227–242 (2010).

Haywood, A. M., Dowsett, H. J., Robinson, M. M., Stoll, D. K., Dolan, A. M., Lunt, D. J., Otto-Bliesner, B., and Chandler, M. A. Pliocene Model Intercomparison Project (PlioMIP): experimental design and boundary conditions (Experiment 2). *Geosci. Model Dev.*, 4, 571–577, (2011).

Iizuka, M., Seki, O., Wilson, D. J. et al. Multiple episodes of ice loss from the Wilkes Subglacial Basin during the Last Interglacial. *Nat. Commun.* 14, 2129 (2023).

Lingle, C. S. & Clark, J. A. A numerical model of interactions between a marine ice sheet and the solid earth: Application to a West Antarctic ice stream. *J. Geophys. Res.* 90, 1100–1114 (1985).

Mengel, M. & Levermann, A. Ice plug prevents irreversible discharge from East Antarctica. *Nat. Clim. Change* 4, 451–455 (2014).

Morlighem, M. et al. Deep glacial troughs and stabilizing ridges unveiled beneath the margins of the Antarctic ice sheet. *Nat. Geosci.* 13, 132–137 (2020).

Rohling, E., Grant, K., Bolshaw, M. et al. Antarctic temperature and global sea level closely coupled over the past five glacial cycles. *Nat. Geosci.* 2, 500–504 (2009).

Stepanek, C., Samakinwa, E., Knorr, G., and Lohmann, G.. Contribution of the coupled atmosphere–ocean–sea ice–vegetation model COSMOS to the PlioMIP2. *Clim. Past*, 16, 2275–2323 (2020).

Sutter, J., Eisen, O., Werner, M., Grosfeld, K., Kleiner, T. & Fischer, H. Limited retreat of the Wilkes Basin ice sheet during the Last Interglacial. *Geophys. Res. Lett.* 47, e2020GL088131 (2020).

Sutter, J., Fischer, H., Grosfeld, K., Karlsson, N. B., Kleiner, T., Van Liefferinge, B., and Eisen, O.. Modelling the Antarctic Ice Sheet across the mid-Pleistocene transition – implications for Oldest Ice, *The Cryosphere*, 13, 2023–2041, (2019).

Wang, Y., Zhao, C., Gladstone, R., Zwinger, T., Galton-Fenzi, B. K. & Christoffersen, P. Sensitivity of the future evolution of the Wilkes Subglacial Basin ice sheet to grounding-line melt parameterizations. The Cryosphere 18, 5117–5137 (2024).